# Reassessing How to Compare and Improve the Calibration of Machine Learning Models

**Muthu Chidambaram & Rong Ge**
Department of Computer Science, Duke University

## Abstract

A machine learning model is calibrated if its predicted probability for an outcome matches the observed frequency for that outcome conditional on the model prediction. This property has become increasingly important as the impact of machine learning models has continued to spread to various domains. As a result, there are now a dizzying number of recent papers on measuring and improving the calibration of (specifically deep learning) models. In this work, we reassess the reporting of calibration metrics in the recent literature. We show that there exist trivial recalibration approaches that can appear seemingly state-of-the-art unless calibration and prediction metrics (i.e. test accuracy) are accompanied by additional generalization metrics such as negative log-likelihood. We then use a calibration-based decomposition of Bregman divergences to develop a new extension to reliability diagrams that jointly visualizes calibration and generalization error, and show how our visualization can be used to detect trade-offs between calibration and generalization. Along the way, we prove novel results regarding the relationship between full calibration error and confidence calibration error for Bregman divergences. We also establish the consistency of the kernel regression estimator for calibration error used in our visualization approach, which generalizes existing consistency results in the literature.

## 1 Introduction

Standard machine learning models are trained to predict probability distributions over a set of possible actions or outcomes. Model-based decision-making is then typically done by using the action or outcome associated with the highest probability, and ideally one would like to interpret the model-predicted probability as a notion of confidence in the predicted action/outcome.

In order for this confidence interpretation to be valid, it is crucial that the predicted probabilities are *calibrated* (Lichtenstein et al., 1982; Dawid, 1982; DeGroot & Fienberg, 1983), or accurately reflect the true frequencies of the outcome conditional on the prediction. As an informal (classic) example, a calibrated weather prediction model would satisfy the property that we observe rain 80% of the time on days for which our model predicted a $0.8$ probability of rain.

As the applications of machine learning models - particularly deep learning models - continue to expand to include high-stakes areas such as medical image diagnoses (Mehrtash et al., 2019; Elmarakeby et al., 2021; Nogales et al., 2021) and self-driving cars (Hu et al., 2023), so too does the importance of having calibrated model probabilities. Unfortunately, the seminal empirical investigation of Guo et al. (2017) demonstrated that deep learning models can be poorly calibrated, largely due to overconfidence.

This observation has led to a number of follow-up works intended to improve model calibration using both training-time (Thulasidasan et al., 2019; Müller et al., 2020; Wang et al., 2021) and post-training methods (Joy et al., 2022; Gupta & Ramdas, 2022). Comparing these proposed improvements, however, is non-trivial due to the fact that the measurement of calibration in practice is itself an active area of research (Nixon et al., 2019; Kumar et al., 2019; Błasiok et al., 2023), and improvements with respect to one calibration measure do not necessarily indicate improvements with respect to another.

Even if we fix a choice of calibration measure, the matter is further complicated by the existence of trivially calibrated models, such as models whose confidence is always their test accuracy (see Section 3). Most works on improving calibration have approached these issues by choosing to report a

number of different calibration metrics along with generalization metrics (e.g. negative log-likelihood or mean-squared error), but these choices vary greatly across works and are always non-exhaustive.

Motivated by this variance in calibration reporting, our work aims to answer the following questions:

- Do there exist any systematic issues in the reporting of calibration in the recent literature?

- Is there a theoretically motivated choice of which calibration and generalization measures to report, as well as an efficient means of estimating and visualizing them jointly?

## 1.1 SUMMARY OF MAIN CONTRIBUTIONS AND TAKEAWAYS

In answering these questions, we make the following contributions.

1. We identify multiple issues (identified in Sections 3, 4.1, and 4.2) in how calibration performance is reported in the context of multi-class classification, with the core issue being that many (even very recent) papers report only some notion(s) of calibration error along with test accuracy. With respect to these metrics, the trivial recalibration strategy of always predicting with the mean confidence obtained from a calibration set outperforms the most popular post-training recalibration methods.

2. Based on this observation, we revisit the use of proper scoring rules for measuring calibration. We apply decomposition results for these scoring rules to show how we can motivate a calibration metric based on our choice of generalization metric, and in special cases how we can do the converse as well. Furthermore, we focus on proper scoring rules that are Bregman divergences and leverage the structure of these divergences to also prove new results regarding the relationship between full calibration and confidence calibration, with our main result being Lemma 4.6 (confidence calibration error is a lower bound for full calibration error).

3. Finally, we propose an extension to reliability diagrams for jointly visualizing calibration and generalization. We also prove the consistency of the estimation approach we use in our new visualization in Theorem 4.9, thereby extending existing estimation results in the literature. In Section 5, we show how our calibration-sharpness diagrams can provide a more granular comparison of recalibration methods when compared to traditional reliability diagrams (Figure 2). Code for our visualizations is available as the Python package sharpcal.

The main takeaway of our results is that **the choice of calibration metric should be motivated by the choice of generalization metric or loss function** and that **both should be reported together** in order to ensure that improvements in model calibration are not at the cost of model sharpness.

## 1.2 RELATED WORK

**Measuring calibration.** Expected calibration error (ECE) is almost certainly the most used calibration metric in the literature, and it is typically estimated using histogram binning (Naeini et al., 2014). Although originally estimated using uniformly spaced bins (Guo et al., 2017), different binning schemes (such as equal mass/quantile-based binning) and debiasing procedures have been proposed to improve estimation of ECE (Nixon et al., 2019; Kumar et al., 2019; Roelofs et al., 2022). As ECE is known to have a number of undesirable properties (e.g. discontinuous in the space of predictors), recent work has focused on modifying ECE to fix these issues (Błasiok & Nakkiran, 2023; Chidambaram et al., 2024).

Alternatives to ECE include proper scoring rules (Gneiting & Raftery, 2007), smooth calibration error (Kakade & Foster, 2008), maximum mean calibration error (Kumar et al., 2018), cumulative plots comparing labels and predicted probabilities (Arrieta-Ibarra et al., 2022), and hypothesis tests for miscalibration (Lee et al., 2022). Several of these approaches are analyzed under the recent theoretical framework of Błasiok et al. (2023), which puts forth a general desiderata for calibration measures.

In practice, most true calibration measures are only efficient for binary classification, and measuring calibration in the multi-class classification setting requires making concessions on the definition of calibration considered. Towards this end, a number of different calibration notions intended to efficiently work in the multi-class setting have been proposed in the literature (Vaicenavicius et al., 2019; Kull et al., 2019; Widmann et al., 2020; Zhao et al., 2021; Gupta & Ramdas, 2022; Gopalan et al., 2024). We elaborate more on binary vs. multi-class calibration in Section 2.

**Improving calibration.** Approaches for improving calibration can largely be categorized into post-training and training-time modifications. In the former category, standard approaches include histogram binning (Mincer & Zarnowitz, 1969; Zadrozny & Elkan, 2001), isotonic regression (Zadrozny & Elkan, 2002), Platt scaling (Platt, 1999), temperature scaling (Guo et al., 2017), and adaptive variants of temperature scaling (Joy et al., 2022). For the latter, data augmentation (Thulasidasan et al., 2019; Müller et al., 2020) and modified versions of the cross-entropy loss (Wang et al., 2021) have been shown to lead to better calibrated models.

**Calibration and proper scoring rules.** Divergences obtained from strictly proper scores such as negative log-likelihood (NLL) and mean-squared error (MSE) often serve as metrics for quantifying generalization. A predictor that is optimal with respect to a strictly proper scoring rule has recovered the ground-truth conditional distribution, and is therefore necessarily calibrated. However, requiring optimality with respect to a proper scoring rule is a stronger condition than requiring calibration, and several works have thus explored how to decompose proper scoring rules into a calibration term and a sharpness/resolution term (Murphy, 1973; Gneiting et al., 2007; Bröcker, 2008; Kull & Flach, 2015; Pohle, 2020). The most related such work to ours is that of Perez-Lebel et al. (2023), which also puts forth the perspective that reporting calibration and accuracy is insufficient; however, they focus on a decomposition involving a grouping loss term whereas we focus on a sharpness term which leads to different visualizations and different takeaways. We provide a more detailed comparison to this work in Appendix E.

## 2 PRELIMINARIES

**Notation.** We use $\overline{\mathbb{R}}$ to denote the extended real line. We use $e_i$ to denote the standard basis vector in $\mathbb{R}^d$ with non-zero $i$ coordinate. Given $n \in \mathbb{N}$, we use $[n]$ to denote the set $\{1, 2, ..., n\}$. For a function $g : \mathbb{R}^n \to \mathbb{R}^m$ we use $g^i(x)$ to denote the $i^{\text{th}}$ coordinate function of $g$. We use $\Delta^{k-1}$ to denote the $(k-1)$-dimensional probability simplex in $\mathbb{R}^k$. For a probability distribution $\pi$ we use $\text{supp}(\pi)$ to denote its support. Additionally, if $\pi$ corresponds to the joint distribution of two random variables $X$ and $Y$ (e.g. data and label), we use $\pi_X$ and $\pi_Y$ to denote the respective marginals and $\pi_{Y|X}$ to denote the regular conditional distribution of $Y$ given $X$. When $Y$ takes values over a finite set, we will identify $\pi_Y$ and $\pi_{Y|X=x}$ with vectors in $\Delta^{k-1}$, and similarly identify $\pi_{Y|X}$ with the corresponding function from $\mathbb{R}^d$ to $\Delta^{k-1}$. For a random variable $X$ that has a density with respect to the Lebesgue measure, we use $p_X(x)$ to denote its density. We use $\sigma(X)$ to denote the sigma-algebra generated by the random variable $X$.

**Calibration.** We restrict our attention to the classification setting, in which there exists a ground-truth distribution $\pi$ on $\mathbb{R}^d \times [k]$ and our goal is to recover the conditional distribution $\pi_{Y|X}$. We say a predictor $g : \mathbb{R}^d \to \Delta^{k-1}$ is *fully calibrated* if it satisfies the condition

$$\mathbb{P}(Y \mid g(X) = v) = v, \tag{2.1}$$

where $\mathbb{P}(Y \mid g(X) = v)$ is the regular conditional distribution of $Y$ given $g(X)$ at $g(X) = v$. Informally, the conditional distribution of the label given our prediction matches our prediction.

In the case where $k = 2$ (binary classification), we can identify $g(X)$ with a single scalar and the condition (2.1) is easy to (approximately) verify as it amounts to estimating a 1-D conditional expectation $\mathbb{E}[Y \mid g(X)]$. On the other hand, when $k$ is large and $g(X)$ has a density with respect to the Lebesgue measure on $\mathbb{R}^k$, the estimation problem becomes significantly more difficult due to the curse of dimensionality.

For such cases, the typical approach taken in practice is to weaken (2.1) to a lower dimensional condition that is easier to work with. To describe such approaches, we adopt the general framework of Gupta & Ramdas (2022). Namely, given a predictor $g$ as before, we consider a classification function $c : \Delta^{k-1} \to [k]$ and a confidence function $h : \Delta^{k-1} \to [0, 1]$ that are used to post-process $g$. For such $c$ and $h$, we can define the weaker notion of *confidence calibration* (Kull et al., 2019) (also sometimes referred to as top-class or top-label calibration):

$$\mathbb{P}(Y = c(g(X)) \mid h(g(X)) = p) = p. \tag{2.2}$$

We refer to the specific case of $c(z) = \text{argmax}_i z_i$ (assuming arbitrary tie-breaking) and $h(z) = \max_i z_i$ as *standard confidence calibration*, since this is usually how confidence calibration appears in the literature. Additionally, to keep notation uncluttered, we will simply use $c(X)$ and $h(X)$

to denote $c(g(X))$ and $h(g(X))$ when $g$ is understood from context in the remainder of the paper. Confidence calibration is by far the most used notion of calibration for multi-class classification in the literature, and we will thus mostly focus our attention on the condition (2.2) moving forward. Alternative notions are discussed in detail in Kull et al. (2019); Gupta & Ramdas (2022); Gopalan et al. (2024).

To evaluate how close a tuple $(c, h, g)$ is to satisfying (2.2), we need a notion of calibration error. As mentioned earlier, the most common choice of calibration error is the Expected Calibration Error (ECE), which is simply the absolute deviation between the left and right-hand sides of (2.2):

$$\mathrm{ECE}_\pi(c, h, g) = \mathbb{E}\left[\left|\mathbb{P}(Y = c(X) \mid h(X) = p) - p\right|\right]. \tag{2.3}$$

This version of ECE is also referred to as the $L^1$ ECE; raising the term inside the expectation to $q > 1$ yields the $L^q$ ECE raised to the power $q$. Intuitively, ECE measures how much our classification accuracy conditioned on our confidence differs from our confidence. In order to estimate the ECE, it is necessary to estimate the conditional probability $\mathbb{P}(Y = c(X) \mid h(X) = p)$. This is done using binning estimators (Naeini et al., 2014; Nixon et al., 2019) and more recently using kernel regression estimators (Popordanoska et al., 2022; Błasiok & Nakkiran, 2023; Chidambaram et al., 2024).

**Proper scoring rules.** Alternatives to ECE for measuring calibration error include divergences obtained from proper scoring rules, which are described in full detail in Gneiting & Raftery (2007). For our purposes, it suffices to consider only the case of scoring rules $\mathcal{S} : \Delta^{k-1} \times \Delta^{k-1} \to \overline{\mathbb{R}}$, in which case $\mathcal{S}$ is called proper if $\mathcal{S}(u, u) \geq \mathcal{S}(u, v)$ for $u \neq v$ and *strictly proper* if equality is only achieved when $v = u$. The divergence $d_\mathcal{S}$ associated with the scoring rule is then defined to be $d_\mathcal{S}(u, v) = \mathcal{S}(u, u) - \mathcal{S}(u, v)$ and is guaranteed to be nonnegative so long as $\mathcal{S}$ is proper.

Perhaps the most popular divergences obtained from (strictly) proper scoring rules in machine learning are the mean squared error (MSE, also referred to as Brier score) and the KL divergence (or just NLL); minimizing either one empirically ensures that $g$ recovers the ground-truth conditional distribution $\pi_{Y|X}$ given sufficiently many samples and appropriate regularity conditions. As mentioned, this is stronger than necessary for achieving (2.2), which is partly why calibration errors such as ECE are appealing as they directly target the condition (2.2).

## 3   Pitfalls in Calibration Reporting

We begin by first pointing out the well-known fact that even the full calibration condition of (2.2) can be achieved by trivial choices of $g$. Indeed, the choice $g(X) = \pi_Y$ satisfies (2.2) despite the fact that $g$ is a constant predictor.

A similar construction is possible for confidence calibration. We can define the confidence function $h$ to be such that $h(z) = \mathbb{P}(Y = \arg\max_i g^i(X))$, i.e. $h$ is a constant equal to the test accuracy of $g$. Taking $c(z) = \arg\max_i z_i$ as in the standard case, we once again have that $\mathrm{ECE}_\pi(c, h, g) = 0$, since:

$$\mathbb{P}(Y = c(X) \mid h(X)) = \mathbb{P}(Y = c(X)) = h(X) \tag{3.1}$$

by construction. Of course in practice, we do not actually know $\mathbb{P}(Y = \arg\max_i g^i(X))$, but we can estimate it on a held-out calibration set and set $h$ to be this estimate. We refer to this approach as **mean replacement recalibarion (MRR)**, and it can be viewed as a special case of the modified histogram binning procedure of Gupta & Ramdas (2022). Since MRR is defined purely through a modified $h$, there is no clear way for getting a vector in $\Delta^{k-1}$ for computing metrics such as NLL or MSE; for such cases, we simply set the non-predicted class probabilities to be $(1 - h(x))/(k - 1)$.

The ramifications of the MRR strategy seem to be under-appreciated in the literature - we have minimized calibration error *without affecting the prediction* of $g$ by using a constant confidence function $h$. By separating the confidence function from the classification function, we have made it so that the test accuracy remains unchanged, so that this kind of trivial confidence behavior is undetectable by just reporting calibration error and accuracy. Even though MRR may seem extreme, one can imagine modifications where we effectively perform the procedure of replacing the predicted confidence with the calibration accuracy while doing something slightly more clever with the other predicted probabilities.

**To what extent is this a problem in the literature?** Although many papers do report multiple calibration and generalization metrics, ECE and accuracy are often the main (or even only) reported

metrics in the main body of these papers. For example, even the influential empirical analysis of Guo et al. (2017) focuses entirely on ECE results in the main paper (NLL results are, however, available in the appendix). This is also largely true of the follow-up re-assessment of deep learning model calibration of Minderer et al. (2021), which focuses on the relationships between test accuracy and ECE for different model families.

For works regarding the improvement of calibration, the work of Thulasidasan et al. (2019) (which motivated the study of Mixup for calibration) reports *only* test accuracy, ECE, and a related overconfidence measure *throughout the whole paper*. Similarly, the works of Zhang et al. (2020) and Wen et al. (2021) analyzing how ensembling strategies can affect calibration also only report test accuracy, ECE, and some ECE variants throughout their whole papers. Calibration has also been an important topic in the context of large language models (LLMs), and initial investigations into the calibration of LLMs for question-answering (Jiang et al., 2020; Desai & Durrett, 2020) also only reported test accuracy and ECE.

The above listing of papers is of course non-exhaustive; we have focused on a handful of influential papers (all have at least 100 citations) to merely point out that reporting (or at least focusing on) only ECE/similar and test accuracy is common practice. That being said, we are not at all calling into question the advances made by the above papers - many of their findings have been repeatedly corroborated by follow-up work. On the other hand, for new work on calibration we strongly advocate for always reporting errors derived from proper scoring rules (i.e. NLL or MSE) alongside pure calibration metrics such as ECE (we put forth a methodology for doing so in Sections 4.1 and 4.2).

**Experiments.** To drive home the point, we revisit the experiments of Guo et al. (2017), but focus on one of the more modern architectural choices of Minderer et al. (2021). Namely, we evaluate the most downloaded pretrained vision transformer (ViT) model[1] (Dosovitskiy et al., 2021; Steiner et al., 2021) available through the `timm` library (Wightman, 2019) with respect to binned ECE, binned ACE (Nixon et al., 2019), SmoothECE (Błasiok & Nakkiran, 2023), negative log-likelihood, and MSE on ImageNet-1K-Val (Russakovsky et al., 2015). We split the data into 10,000 calibration samples (20% split) and 40,000 test samples, and compare the unmodified test performance to temperature scaling (TS), histogram binning (HB), isotonic regression (IR), and our proposed MRR. We follow the same TS implementation as Guo et al. (2017) and also use 15 bins for the binning estimators (ECE, ACE, HB) to be comparable to the results in their work. The results shown in Table 1 are not sensitive to the choice of model; we show similar results for several other popular `timm` models in Appendix F.

| Model | Test Accuracy | ECE | ACE | SmoothECE | NLL | MSE/Brier Score |
|---|---|---|---|---|---|---|
| ViT (Baseline) | **85.14** | 9.42 | 9.44 | 9.44 | 65.35 | 22.73 |
| ViT + TS | **85.14** | 2.59 | 5.46 | 2.60 | **56.83** | **21.95** |
| ViT + HB | 82.03 | 5.06 | 11.97 | 4.03 | $\infty$ | 27.17 |
| ViT + IR | 83.82 | 4.53 | 8.13 | 4.48 | $\infty$ | 24.13 |
| ViT + MRR | **85.14** | **0.18** | **0.18** | **0.18** | 144.66 | 27.51 |

Table 1: Comparison of different recalibration methods applied to pretrained ViT model on ImageNet-1K-Val. All metrics are scaled by a factor of 100 for readability. Histogram binning and isotonic regression have unbounded NLL as they can predict zero probability values for some classes.

*Remark* 3.1. It is possible to detect the specific instances of trivial calibration discussed above using alternatives to proper scoring rules, such as ROC-AUC applied to the confidence calibration problem. However, the confidence calibration problem can behave very differently than the overall multi-class classification problem (illustrated in Section 4.1), and typically it is very efficient to compute the proper scoring rules applied to the multi-class problem directly.

## 4 MAIN THEORETICAL RESULTS

As can be seen in Table 1, the MRR strategy dominates in terms of all of the pure calibration metrics, but leads to large jumps in NLL and MSE. Clearly either NLL or MSE alone would have been sufficient to detect the trivial behavior of MRR. We now address the more general question: which

---

[1]The model card is available at: `https://huggingface.co/timm/vit_base_patch16_224.augreg2_in21k_ft_in1k`.

generalization metrics should we report to prevent misleading calibration results? Our answer relies on the fact that such metrics are usually *Bregman divergences*. Our focus on Bregman divergences is due to their convenient theoretical properties; in particular, they are defined in terms of convex functions and we can thus bring to bear the machinery of convex analysis.

We first review the definitions. Let us use $\phi : \Delta^{k-1} \to \overline{\mathbb{R}}$ to denote a continuously differentiable $(C^1)$, strictly convex function. We recall that the Bregman divergence $d_\phi$ associated with $\phi$ is defined to be:

$$d_\phi(x, y) = \phi(x) - \phi(y) - \nabla\phi(y)^\top (x - y). \tag{4.1}$$

In many machine learning tasks the goal is to minimize the expectation of a conditional Bregman divergence; for example, in multi-class classification we take $\phi(x) = \sum_i x_i \log x_i$ to obtain the KL divergence and then attempt to minimize $\mathbb{E}[d_\phi(\pi_{Y|X}, g(X))]$ from samples. Bregman divergences are closely related with strictly proper scoring rules, and we can leverage the fact that such rules can be decomposed into a calibration error and a "sharpness" term to obtain a similar decomposition for Bregman divergences.

**Lemma 4.1.** *[Bregman Divergence Decomposition] For $d_\phi$ defined as in* (4.1) *and two integrable random variables $Z$ and $X$ defined on the same probability space, it follows that:*

$$\mathbb{E}[d_\phi(Z, X)] = \underbrace{\mathbb{E}[\phi(Z)] - \mathbb{E}[\phi(\mathbb{E}[Z \mid X])]}_{\text{Sharpness Gap}} + \underbrace{\mathbb{E}[d_\phi(\mathbb{E}[Z \mid X], X)]}_{\text{Calibration Error}}. \tag{4.2}$$

More general versions of Lemma 4.1 have appeared in Bröcker (2009); Kull & Flach (2015); Pohle (2020); Gruber & Buettner (2024), but we present a version for Bregman divergences as we will leverage the structure of (4.1) to prove our new results in the subsequent sections. We provide a self-contained proof of Lemma 4.1 in Appendix A.

Some intuition for the sharpness gap term is provided in the next two propositions, the first of which appeared in Pohle (2020) in a different form. The second follows easily from the first and Lemma 4.1, but has not appeared elsewhere to our knowledge. Intuitively, the propositions show that a smaller sharpness gap corresponds to a predictor that is more "fine-grained", and in that sense the constant calibrated predictor that we introduced in the previous subsection is the worst possible fully calibrated predictor. The proofs of both propositions can once again be found in Appendix A.

**Proposition 4.2.** *For $g_1, g_2 : \mathbb{R}^d \to \Delta^{k-1}$ and $\pi$ the joint distribution of data and label $(X, Y)$ on $\mathbb{R}^d \times [k]$, if $\sigma(g_1(X)) \subseteq \sigma(g_2(X))$, then:*

$$\mathbb{E}[\phi(\mathbb{E}[\pi_{Y|X} \mid g_1(X)])] \leq \mathbb{E}[\phi(\mathbb{E}[\pi_{Y|X} \mid g_2(X)])]. \tag{4.3}$$

**Proposition 4.3.** *Let $\pi$ correspond to the joint distribution of data and label $(X, Y)$ on $\mathbb{R}^d \times [k]$. Then every predictor $g$ that is calibrated in the sense of* (2.1) *satisfies:*

$$\mathbb{E}[d_\phi(\pi_{Y|X}, g(X))] \leq \mathbb{E}[d_\phi(\pi_{Y|X}, \pi_Y)]. \tag{4.4}$$

The main takeaway from Lemma 4.1 is that when we choose to optimize a Bregman divergence, we implicitly also have a notion of calibration error that we are minimizing via (4.2). This provides a **practical workflow** to answer our motivating question of which metrics to report together: we first fix a Bregman divergence for our notion of generalization, and then *report the same divergence* but applied to the expected label conditioned on the predicted probabilities. We can also ask about the other direction – if we choose a calibration error, is there a natural Bregman divergence associated with it?

The answer is: not always. In fact, in considering this question, we can motivate a choice of $L^q$ ECE. The next result shows that in the framework of Bregman divergences the "right" choice of ECE is the $L^2$ ECE, since it is the conditional expectation of the Brier score.

**Proposition 4.4.** *Let $\pi$ correspond to the joint distribution of data and label $(X, Y)$ on $\mathbb{R}^d \times [k]$, and define the $L^q$ (for integers $q \geq 1$) ECE of a function $g : \mathbb{R}^d \to \Delta^{k-1}$ to be $\mathbb{E}\left[\|\mathbb{E}[Y \mid g(X)] - g(X)\|_q^q\right]$. Then only the $L^2$ ECE is induced by a Bregman divergence in the sense of Lemma 4.1.*

The proof is a simple consequence of characterizations provided in Boissonnat et al. (2010) (which we recall in Appendix A).

## 4.1 Confidence Calibration With Bregman Divergences

So far, we have leveraged existing ideas in the literature on proper scoring rules to motivate a joint choice of generalization and calibration error. However, as we alluded to earlier, when considering high-dimensional classification problems it is not feasible to efficiently estimate the conditional expectation required for calibration error. The main result of this section is to show that we can replace the full calibration error in (4.2) with confidence calibration error and still obtain a valid lower bound, which provides justification for the reporting of confidence calibration at least in the context of Bregman divergences.

In order to discuss confidence calibration, we need a 1-D counterpart $\varphi : [0,1] \to \overline{\mathbb{R}}$ to $\phi$. For our cases of interest, the choice of $\varphi$ is obvious; for example, when $\phi(x) = \|x\|^2$ (i.e. $d_\phi$ is Brier score), $\varphi(x) = x^2$. Recalling that in standard confidence calibration we have $c(g(X)) = \mathrm{argmax}_i g^i(X)$ and $h(g(X)) = \max_i g^i(X)$, our goal is then to investigate the relationship between $\mathbb{E}[d_\varphi(\mathbb{E}[\mathbb{1}_{Y=c(X)} \mid h(X)], h(X))]$ and $\mathbb{E}[d_\phi(\mathbb{E}[\pi_{Y|X} \mid g(X)], g(X))]$.

Prior to doing so, let us pause to stress the fact that the confidence calibration framework should be applied only to the calibration error and not to the entire loss. In other words, we should report the full Bregman divergence $\mathbb{E}[d_\phi(\pi_{Y|X}, g(X))]$ in addition to the calibration error, not the 1-D Bregman divergence $\mathbb{E}[d_\varphi(\mathbb{1}_{Y=c(X)}, h(X))]$. Reporting of the latter is **another pitfall** and has appeared in the recent literature on surveying confidence elicitation strategies for LLMs (Tian et al., 2023). We show in the next proposition that the 1-D loss can have very different behavior when compared to the full loss for the specific case of MSE/Brier score.

**Proposition 4.5.** *Suppose $\pi$ satisfies the property that there exists $y^* : \Delta^{k-1} \to [k]$ such that $\pi_{Y|X=x}(y^*(x)) = 1$ for $\pi_X$-a.e. $x$. Now consider any function $\tilde{y} : \Delta^{k-1} \to [k]$ satisfying $\tilde{y}(x) \neq y^*(x)$ for $\pi_X$-a.e. $x$. Then the "always wrong" predictor defined as $g^{\tilde{y}(x)}(x) = 1/k + \epsilon$ and $g^y(x) = 1/k - \epsilon/(k-1)$ for all $y \neq \tilde{y}(x)$ with $\epsilon = O(1/k)$ satisfies*

$$\mathbb{E}\left[\left(\mathbb{1}_{Y=c(X)} - h(X)\right)^2\right] = O(1/k) \quad but \quad \mathbb{E}\left[\left\|\pi_{Y|X} - g(X)\right\|^2\right] = \Omega(1) \qquad (4.5)$$

*for the standard confidence calibration choices $c(z) = \mathrm{argmax}_i z_i$ and $h(z) = \max_i z_i$.*

The proof follows from directly evaluating (4.5), and can be found in Appendix B. Proposition 4.5 shows that the loss $d_\varphi$ incurred on the confidence calibration problem can be arbitrarily small for a *predictor that is always wrong* despite such a predictor having true loss on the multi-class classification problem bounded below by a constant. We now return to our original goal, which was showing that $\mathbb{E}[d_\varphi(\mathbb{E}[\mathbb{1}_{Y=c(X)} \mid h(X)], h(X))]$ is a lower bound for $\mathbb{E}[d_\phi(\mathbb{E}[\pi_{Y|X} \mid g(X)], g(X))]$ under the condition that $\phi$ dominates $\varphi$ applied to any individual coordinate pair (for example, this is easily seen to be the case for Brier score).

**Lemma 4.6.** *For any $C^1$, convex functions $\phi : \Delta^{k-1} \to \overline{\mathbb{R}}$ and $\varphi : [0,1] \to \overline{\mathbb{R}}$ such that their Bregman divergences satisfy $d_\phi(x,y) \geq d_\varphi(x_i, y_i)$ for all $i \in [k]$, it follows (with probability 1) that:*

$$\mathbb{E}[d_\phi(\mathbb{E}[\pi_{Y|X} \mid g(X)], g(X)) \mid h(X)] \geq d_\varphi(\mathbb{E}[\mathbb{1}_{Y=c(X)} \mid h(X)], h(X)). \qquad (4.6)$$

**Corollary 4.7.** *The inequality* (4.6) *holds for the choices $\phi(x) = \sum_i x_i \log x_i$ (KL divergence) and $\varphi(x) = x \log x + (1-x)\log(1-x)$ (logistic loss).*

The proof of Lemma 4.6 relies on a careful analysis of the conditional expectation terms, and can also be found in Appendix B. Corollary 4.7 follows from the data processing inequality.

## 4.2 Calibration-Sharpness Diagrams

To summarize our observations thus far, we have recommended that calibration should be compared using the calibration metric associated with the Bregman divergence for which the models were optimized, and that both should be reported together. It remains to address how to similarly modify the visualization of (confidence) calibration, which is typically done via reliability diagrams that plot $\mathbb{E}[\mathbb{1}_{Y=c(X)} \mid h(X)]$ for different values of $h(X)$. Our goals are three-fold: we want a visualization that is efficient to compute, makes it obvious whether one model is better calibrated and/or has better overall divergence than another model, and allows one to pinpoint the regions of the predicted probability space where model performance is weak. While there are a few recent works that try to

make progress on these goals, we are not aware of a solution presented as a single diagram satisfying our desiderata (Dimitriadis et al., 2023; Gneiting & Resin, 2023).

The approach we take is to augment the plot of $\mathbb{E}[\mathbb{1}_{Y=c(X)} \mid h(X)]$ with a surrounding band whose length $\rho(h(X))$ corresponds to the sharpness gap from Lemma 4.1 at $h(X)$, which we refer to as **calibration-sharpness diagrams**. For convenience, we will use $\rho(X)$ in place of $\rho(h(X))$ when $h$ is understood. The function $\rho : [0, 1] \to \overline{\mathbb{R}}$ is the **pointwise sharpness gap**, and is formally defined as:

$$\rho(p) = \mathbb{E}\left[d_\phi(\pi_{Y|X}, g(X)) \mid h(X) = p\right] - d_\varphi\left(\mathbb{E}[\mathbb{1}_{Y=c(X)} \mid h(X) = p], p\right), \qquad (4.7)$$

where as in the previous subsection $d_\phi$ is the full Bregman divergence and $d_\varphi$ is its 1-D counterpart. Note that in (4.7) we are conditioning $d_\phi(\pi_{Y|X}, g(X))$ on $h(X)$, since we want a value for the divergence at different confidence levels $h(X)$. The key property of $\rho$ which allows it to be used as a visualization aid is the following.

**Proposition 4.8.** *Suppose that $\phi$ and $\varphi$ satisfy the conditions of Lemma 4.6. Then the function $\rho$ as defined in (4.7) is nonnegative.*

The proof of Proposition 4.8 (which can be found in Appendix 4.2) follows from applying Lemmas 4.1 and Lemma 4.6. The function $\rho$ shows how we trade off calibration with generalization at any confidence level $p$, and we can compute $\rho$ efficiently as it only requires estimating 1-D conditional expectations.

Our estimation of $\rho$ relies on Nadaraya-Watson kernel regression (Nadaraya, 1964; Watson, 1964) with a kernel $K$ and bandwidth hyperparameter $\sigma$ following the estimation approaches of Błasiok & Nakkiran (2023) and Chidambaram et al. (2024). We recall that given $n$ data points $(x_i, y_i) \in \mathbb{R} \times \mathbb{R}$ the kernel regression estimator of $\mathbb{E}[Y \mid X = x]$ is defined as:

$$\hat{m}_\sigma(x) = \frac{\sum_{i=1}^n \frac{1}{\sigma} K\left(\frac{x-x_i}{\sigma}\right)}{\sum_{i=1}^n \frac{1}{\sigma} K\left(\frac{x-x_i}{\sigma}\right)} = \frac{\sum_{i=1}^n K_\sigma(x - x_i)}{\sum_{i=1}^n K_\sigma(x - x_i)}. \qquad (4.8)$$

Previous work (Zhang et al., 2020; Popordanoska et al., 2022; Chidambaram et al., 2024) has shown that we can obtain consistent estimators of $L^p$ ECE using (4.8). We extend these results to calibration errors derived from Bregman divergences. Due to the technical nature of some of the assumptions necessary on $K_\sigma$ for this result, we defer a proper presentation to Section 4.2 and present only an informal version of the result below. The proof relies on uniform convergence results for kernel regression applied to the individual terms in (4.1).

**Theorem 4.9.** *Under suitable conditions on $K$ and the sequence of bandwidths $\sigma_n$, the plugin estimate using $\hat{m}_{\sigma_n}(h(x))$ of the confidence calibration error with respect to $d_\varphi$ is consistent so long as $h(X)$ has a density bounded away from 0 and $\mathbb{E}[\mathbb{1}_{Y=c(X)} \mid h(X)]$ is continuous.*

**Constructing and reading calibration-sharpness diagrams.** Using the above, we construct a diagram with three components: the calibration curve, the pointwise sharpness gap $\rho(h(X))$, and the density plot $\hat{p}_{h(X)}$. Example diagrams are shown in Figures 1 and 2 and interpreted in detail in Section 5. The calibration curve corresponds to $\mathbb{E}[\mathbb{1}_{Y=c(x)} \mid h(x)]$ plotted at uniformly spaced values of $h(x) \in [0, 1]$, and shows the conditional model accuracy vs. its confidence. A perfectly confidence-calibrated model satisfying $\mathbb{E}[\mathbb{1}_{Y=c(x)} \mid h(x)] = h(x)$ is visualized as a dashed line, and any deviations from this dashed line can be interpreted as weaknesses in model calibration.

The sharpness gap is plotted as a band around the calibration curve, and the size of the band corresponds to (4.7) evaluated at each $h(x)$. *A larger band is worse*, as it means a bigger sharpness gap at a predicted confidence $h(x)$, which implies a worse generalization error at that particular confidence level. We also plot a kernel density estimate $\hat{p}_{h(X)}$ of the predicted confidences – this is intended to show where the model confidences concentrate. A density peak at $h(x) \approx 0.8$, for example, indicates that the model typically predicts around 80% confidence for its predicted class.

We point out that another **common pitfall** in several prior works is to omit this kind of density plot when discussing calibration, since it does not matter if a model is perfectly calibrated in a region of its confidence space that it rarely predicts. Lastly, we also overlay the estimated values of the confidence calibration error ($d_{\varphi,\mathrm{CAL}}$) and the full divergence ($d_{\phi,\mathrm{TOT}}$) on all plots, so as to give a sense of the full generalization error of the model as well as the part attributable to calibration. More precise estimation details for each component of the diagram are provided in Appendix D.

## 5 EXPERIMENTS

To illustrate the benefits of calibration-sharpness diagrams, we first use them to revisit the experiments of Section 3 and then provide a direct comparison to the standard approach of reliability diagrams. For all of the experiments in this section, we focus on the ViT model of Section 3 for consistency and use Brier score/MSE since we will be comparing to ECE-based visualization methods in Section 5.2. We provide visualizations for other models and datasets in Appendix F. The kernel regressions estimates for the visualized components are computed using a Gaussian kernel with bandwidth $\sigma = 0.05$; we discuss different choices of kernel and bandwidth in Appendix H. All of our experiments were done in PyTorch (Paszke et al., 2019) on a single A5000 GPU.

### 5.1 REVISITING MRR EXPERIMENTS

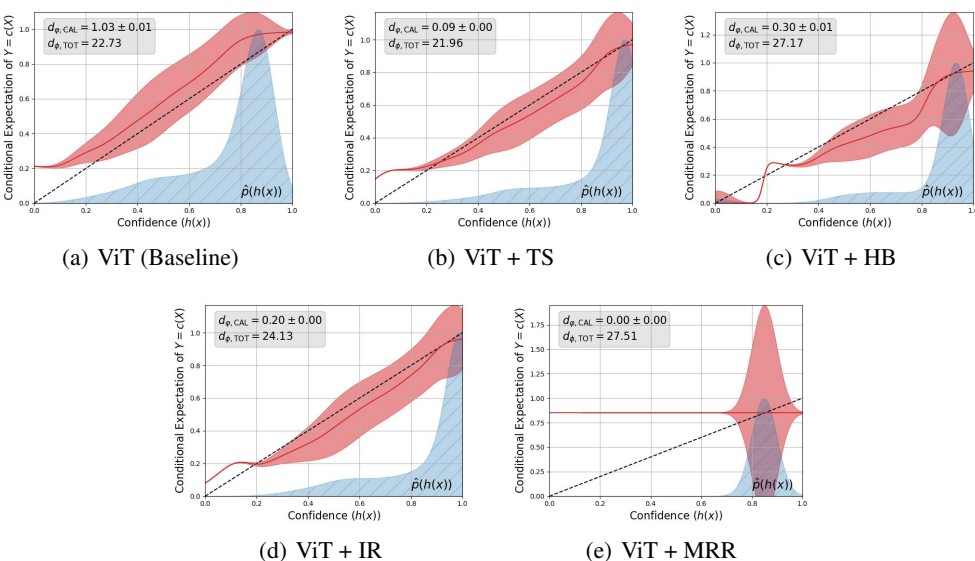

(a) ViT (Baseline)        (b) ViT + TS        (c) ViT + HB

(d) ViT + IR        (e) ViT + MRR

Figure 1: Proposed calibration-sharpness diagrams using MSE/Brier score for the experiments of Section 3. Y-axes are not aligned here due to the large sharpness gap of MRR.

Figure 1 shows the calibration-sharpness diagrams (using $d_\phi(x, y) = \|x - y\|^2$) for the ViT recalibration approaches in Table 1. The first thing to notice is that the calibration-sharpness diagram immediately reveals the weaknesses of both MRR and histogram binning - these methods using a small fixed set of predicted confidences (a single confidence in the case of MRR), and as a result incur large sharpness gaps as shown by the red shaded regions.

Additionally, when comparing the better recalibration approaches of temperature scaling and isotonic regression to the baseline model, we see that these algorithms have actually (perhaps counterintuitively) skewed the predicted confidence distributions towards *even higher probabilities* (as shown by the overlayed density plots of $\hat{p}_{h(X)}(h(x))$). Moreover, we see from the pointwise sharpness gap visualizations that the calibration improvements from isotonic regression (and to a lesser extent, temperature scaling) have come at the cost of some increased sharpness gaps. This can be expected from the fact that both methods led to larger predicted confidences, so mistakes incur larger errors.

### 5.2 COMPARISON TO RELIABILITY DIAGRAMS

The standard approach taken by many empirical papers analyzing confidence calibration is to report *reliability diagrams*. These diagrams correspond to histogram binning estimators of the calibration error – namely, we sort the predicted confidences $h(x)$ into uniformly spaced buckets in the interval $[0, 1]$ and plot a bar corresponding to the average accuracy in each bucket. Similar to the dashed line in our calibration-sharpness diagrams, perfectly confidence calibrated models would have bars that line up with the average confidence of each bucket (i.e. the height of the bar for the $[0, 0.1]$ bucket should be somewhere in $[0, 0.1]$).

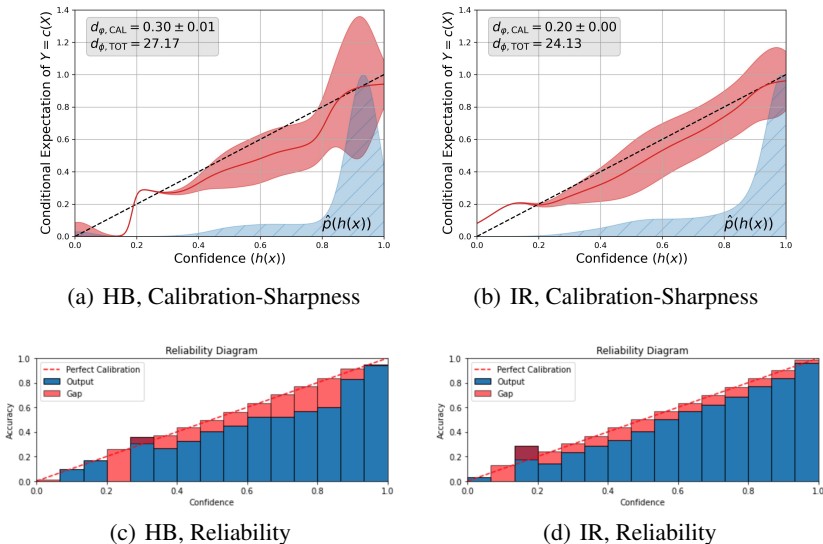

(a) HB, Calibration-Sharpness          (b) IR, Calibration-Sharpness

(c) HB, Reliability                (d) IR, Reliability

Figure 2: Comparing calibration-sharpness diagrams to reliability diagrams using histogram binning and isotonic regression. Y-axes of HB and IR are aligned for more direct comparison.

Figure 2 shows reliability diagrams (using 15 bins, as was done in Guo et al. (2017)) for the histogram binning (HB) and isotonic regression (IR) recalibration approaches juxtaposed with axis-aligned calibration-sharpness diagrams from Figure 1. The reliability diagrams suggest that IR leads to better calibration than HB in the confidence range around $0.2$ and in $[0.5, 0.9]$. However, these diagrams cannot be used to compare generalization and also do not give a sense of where predicted confidences concentrate.

On the other hand, the calibration-sharpness diagrams allow us to compare generalization while providing a more granular view of calibration performance as well. They show that (1) while the calibration of HB is worse than IR, the confidence region around $0.2$ is not a major problem for HB since there is no confidence density there and (2) that IR has better generalization performance (with respect to Brier score) due to smaller sharpness gaps.

Our work is not the first to point out these drawbacks of reliability diagrams for comparing calibration; the work of Błasiok & Nakkiran (2023) on SmoothECE also discusses such issues and puts forth a visualization that includes density plots combined with a kernel regression estimate of the calibration error as opposed to a binning estimate. However, their approach does not include a sharpness component, which in this instance is crucial for comparing HB and IR since this is where HB really loses out. We compare to the SmoothECE visualization approach in Appendix G.

## 6   CONCLUSION

In this work, we have reviewed the reporting of calibration metrics in the recent literature and identified several pitfalls that appear even in well-known works. To address the pointed out issues, we have proposed a reporting methodology based on proper scoring rules that are Bregman divergences, and shown in both theory and practice how our approach fits alongside the popular notion of confidence calibration in the literature. In developing our methodology, we have also introduced a new extension of reliability diagrams for visualizing calibration.

The core tenets we hope to emphasize through both our theory and experiments are: report test accuracy and calibration error alongside the appropriate generalization metric, use a visualization of calibration that shows the density of predicted confidences, and be aware of whether improvements in calibration in some regions of predicted confidence come at the cost of sharpness/generalization. We believe there is still much work to be done on addressing how to compare and visualize calibration/generalization metrics, and we hope future work continues to push in this direction.

## ETHICS STATEMENT

While calibration has become an increasingly important topic given the use of machine learning models for decision-making, our work concerns only the theory and practice of reporting calibration. As a result, we do not anticipate any negative societal impacts or significant ethical considerations as consequences of this work.

## REPRODUCIBILITY STATEMENT

Full proofs of all results in the main paper can be found in the appendices following the references. The code necessary to recreate the experiments in the main paper as well as the experiments in the appendices can be found in the supplementary material, as well as in the Python package sharpcal.

## ACKNOWLEDGMENTS

Rong Ge and Muthu Chidambaram were supported by NSF Award DMS-2031849 and CCF-1845171 (CAREER) during the completion of this work.

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

CONTENTS

## A BACKGROUND RESULTS

**Lemma 4.1.** *[Bregman Divergence Decomposition] For $d_\phi$ defined as in* (4.1) *and two integrable random variables $Z$ and $X$ defined on the same probability space, it follows that:*

$$\mathbb{E}[d_\phi(Z,X)] = \underbrace{\mathbb{E}[\phi(Z)] - \mathbb{E}[\phi(\mathbb{E}[Z \mid X])]}_{\textit{Sharpness Gap}} + \underbrace{\mathbb{E}[d_\phi(\mathbb{E}[Z \mid X], X)]}_{\textit{Calibration Error}}. \tag{4.2}$$

*Proof.* We observe that by (4.1), we have:

$$\mathbb{E}[d_\phi(\mathbb{E}[Z \mid X], X)] = \mathbb{E}[\phi(\mathbb{E}[Z \mid X])] - \mathbb{E}[\phi(X)] - \mathbb{E}[\nabla\phi(X)^\top(\mathbb{E}[Z \mid X] - X)]$$
$$= \mathbb{E}[\phi(\mathbb{E}[Z \mid X])] - \mathbb{E}[\phi(X)] - \mathbb{E}[\nabla\phi(X)^\top(Z - X)]. \tag{A.1}$$

Where the last line follows from the fact that $\mathbb{E}[Y\mathbb{E}[Z \mid X]] = \mathbb{E}[YZ]$ for any $X$-measurable random variable $Y$. Equation (4.2) then immediately follows from applying definition (4.1). $\qquad\square$

**Proposition 4.2.** *For $g_1, g_2 : \mathbb{R}^d \to \Delta^{k-1}$ and $\pi$ the joint distribution of data and label $(X, Y)$ on $\mathbb{R}^d \times [k]$, if $\sigma(g_1(X)) \subseteq \sigma(g_2(X))$, then:*

$$\mathbb{E}[\phi(\mathbb{E}[\pi_{Y\mid X} \mid g_1(X)])] \leq \mathbb{E}[\phi(\mathbb{E}[\pi_{Y\mid X} \mid g_2(X)])]. \tag{4.3}$$

*Proof.* By Jensen's inequality and the tower property of conditional expectation, we have:

$$\mathbb{E}[\phi(\mathbb{E}[\pi_{Y\mid X} \mid g_2(X)])] = \mathbb{E}[\mathbb{E}[\phi(\mathbb{E}[\pi_{Y\mid X} \mid g_2(X)]) \mid g_1(X)]]$$
$$\geq \mathbb{E}[\phi(\mathbb{E}[\mathbb{E}[\pi_{Y\mid X} \mid g_2(X)] \mid g_1(X)])]$$
$$= \mathbb{E}[\phi(\mathbb{E}[\pi_{Y\mid X} \mid g_1(X)])].$$

$\qquad\square$

**Proposition 4.3.** *Let $\pi$ correspond to the joint distribution of data and label $(X, Y)$ on $\mathbb{R}^d \times [k]$. Then every predictor $g$ that is calibrated in the sense of* (2.1) *satisfies:*

$$\mathbb{E}[d_\phi(\pi_{Y\mid X}, g(X))] \leq \mathbb{E}[d_\phi(\pi_{Y\mid X}, \pi_Y)]. \tag{4.4}$$

*Proof.* Firstly, we observe that for any calibrated predictor we have $\mathbb{E}[d_\phi(\mathbb{E}[\pi_{Y\mid X} \mid g(X)], g(X)] = 0$. Next, we note that for every $g : \mathbb{R}^d \to \Delta^{k-1}$, $g(X)$ is measurable with respect to the $\sigma$-algebra generated by the constant predictor $\pi_Y$ (since this is the trivial $\sigma$-algebra). The inequality (4.4) then follows by applying Proposition 4.2 to the decomposition from Lemma 4.1. $\qquad\square$

**Proposition 4.4.** *Let $\pi$ correspond to the joint distribution of data and label $(X, Y)$ on $\mathbb{R}^d \times [k]$, and define the $L^q$ (for integers $q \geq 1$) ECE of a function $g : \mathbb{R}^d \to \Delta^{k-1}$ to be $\mathbb{E}\left[\|\mathbb{E}[Y \mid g(X)] - g(X)\|_q^q\right]$. Then only the $L^2$ ECE is induced by a Bregman divergence in the sense of Lemma 4.1.*

*Proof.* We first note that the case of $q = 1$ is not a Bregman divergence since Bregman divergences are differentiable in their first argument. For $q > 1$, Lemma 2 in Boissonnat et al. (2010) shows that a twice differentiable function $\phi$ induces a symmetric Bregman divergence $d_\phi$ iff the Hessian of $\phi$ is constant. This implies that for any $L^q$ norm (raised to the $q$ power) to be a Bregman divergence, differentiating it twice in the first argument must yield a constant. It is easy to see that this is only the case for $q = 2$, and the result follows. $\qquad\square$

## B  PROOFS FOR SECTION 4.1

**Proposition 4.5.** *Suppose $\pi$ satisfies the property that there exists $y^* : \Delta^{k-1} \to [k]$ such that $\pi_{Y|X=x}(y^*(x)) = 1$ for $\pi_X$-a.e. $x$. Now consider any function $\tilde{y} : \Delta^{k-1} \to [k]$ satisfying $\tilde{y}(x) \neq y^*(x)$ for $\pi_X$-a.e. $x$. Then the "always wrong" predictor defined as $g^{\tilde{y}(x)}(x) = 1/k + \epsilon$ and $g^y(x) = 1/k - \epsilon/(k-1)$ for all $y \neq \tilde{y}(x)$ with $\epsilon = O(1/k)$ satisfies*

$$\mathbb{E}\left[\left(\mathbb{1}_{Y=c(X)} - h(X)\right)^2\right] = O(1/k) \quad but \quad \mathbb{E}\left[\left\|\pi_{Y|X} - g(X)\right\|^2\right] = \Omega(1) \tag{4.5}$$

*for the standard confidence calibration choices $c(z) = \operatorname{argmax}_i z_i$ and $h(z) = \max_i z_i$.*

*Proof.* Observing that $\mathbb{1}_{Y=c(X)} = 0$ and $h(X) = 1/k + \epsilon$ gives the first part of (4.5). On the other hand, the leading term of $\mathbb{E}[\|\pi_{Y|X} - g(X)\|^2]$ is $(1 - 1/k + \epsilon/(k-1))^2$, which is $\Omega(1)$. □

**Lemma 4.6.** *For any $C^1$, convex functions $\phi : \Delta^{k-1} \to \overline{\mathbb{R}}$ and $\varphi : [0,1] \to \overline{\mathbb{R}}$ such that their Bregman divergences satisfy $d_\phi(x,y) \geq d_\varphi(x_i, y_i)$ for all $i \in [k]$, it follows (with probability 1) that:*

$$\mathbb{E}[d_\phi(\mathbb{E}[\pi_{Y|X} \mid g(X)], g(X)) \mid h(X)] \geq d_\varphi(\mathbb{E}[\mathbb{1}_{Y=c(X)} \mid h(X)], h(X)). \tag{4.6}$$

*Proof.* Firstly, recalling that $e_{c(X)}$ denotes the basis vector in $\mathbb{R}^k$ with non-zero index $c(X)$,

$$\begin{aligned}
\mathbb{E}[\mathbb{1}_{Y=c(X)} \mid g(X)] &= \mathbb{E}[\mathbb{E}[\mathbb{1}_{Y=c(X)} \mid X] \mid g(X)] \\
&= \mathbb{E}[e_{c(X)}^\top \pi_{Y|X} \mid g(X)] \\
&= e_{c(X)}^\top \mathbb{E}[\pi_{Y|X} \mid g(X)],
\end{aligned} \tag{B.1}$$

where the last line follows due to the measurability of $e_{c(X)}$ with respect to $g(X)$. By the assumptions on $d_\phi$ and $d_\varphi$ and the fact that $h(X) = e_{c(X)}^\top g(X)$, we have:

$$\begin{aligned}
d_\phi(\mathbb{E}[\pi_{Y|X} \mid g(X)], g(X)) &\geq d_\varphi(\mathbb{E}[\pi_{Y|X} \mid g(X)]_{c(X)}, g(X)_{c(X)}) \\
&= d_\varphi(e_{c(X)}^\top \mathbb{E}[\pi_{Y|X} \mid g(X)], e_{c(X)}^\top g(X)) \\
&= d_\varphi(\mathbb{E}[\mathbb{1}_{Y=c(X)} \mid g(X)], h(X)).
\end{aligned} \tag{B.2}$$

The desired (4.6) then follows by noting that

$$\begin{aligned}
\mathbb{E}[d_\varphi(\mathbb{E}[\mathbb{1}_{Y=c(X)} \mid g(X)], h(X)) \mid h(X)] &= \mathbb{E}[\varphi(\mathbb{E}[\mathbb{1}_{Y=c(X)} \mid g(X)]) \mid h(X)] - \varphi(h(X)) \\
&\quad - \mathbb{E}[\varphi'(h(X))(\mathbb{E}[\mathbb{1}_{Y=c(X)} \mid g(X)] - h(X)) \mid h(X)] \\
&\geq \varphi(\mathbb{E}[\mathbb{1}_{Y=c(X)} \mid h(X)]) - \varphi(h(X)) \\
&\quad - \varphi'(h(X))(\mathbb{E}[\mathbb{1}_{Y=c(X)} \mid h(X)] - h(X))
\end{aligned} \tag{B.3}$$

due to Jensen's inequality and $h(X)$-measurability of $\varphi'(h(X))$. □

**Corollary 4.7.** *The inequality (4.6) holds for the choices $\phi(x) = \sum_i x_i \log x_i$ (KL divergence) and $\varphi(x) = x \log x + (1-x) \log(1-x)$ (logistic loss).*

*Proof.* We need only show that $d_\phi(x,y) \geq d_\varphi(x_i, y_i)$ for the choices of KL divergence and logistic loss. Let us as usual identify $x$ and $y$ with vectors in $\mathbb{R}^k$ corresponding to distributions in $\Delta^{k-1}$. Then, fixing an $i \in [k]$, the map $f(x) = \text{Bernoulli}(x_i)$ is a Markov kernel, and by the data-processing inequality we have $d_\phi(x,y) \geq d_\phi(f(x), f(y))$. Since $d_\phi(f(x), f(y)) = d_\varphi(x_i, y_i)$, the result follows. □

## C  PROOFS FOR SECTION 4.2

**Proposition 4.8.** *Suppose that $\phi$ and $\varphi$ satisfy the conditions of Lemma 4.6. Then the function $\rho$ as defined in (4.7) is nonnegative.*

*Proof.* From Lemma 4.1, we have:

$$\mathbb{E}\left[d_\phi(\pi_{Y|X}, g(X)) \mid h(X) = p\right] \geq \mathbb{E}\left[d_\phi(\mathbb{E}[\pi_{Y|X} \mid g(X)], g(X)) \mid h(X) = p\right]. \tag{C.1}$$

Applying Lemma 4.6 then yields the result. □

To state and prove a formal version of Theorem 4.9, we need a result of Devroye (1978). We first state the necessary kernel and distributional assumptions for this result. We state everything in the generality of $\mathbb{R}^d$ as that is how they are presented in Devroye (1978), but for our particular result we only need the 1-D analogues.

**Assumption C.1** (Kernel Assumptions). We assume the kernel $K : \mathbb{R}^d \to \mathbb{R}$ satisfies the following:

1. $K(x) \in [0, K^*]$ for some $K^* < \infty$.

2. $K(x) = L(\|x\|)$ for some nonincreasing function $L$.

3. $\lim_{u \to \infty} uL(u) = 0$.

4. $L(u^*) > 0$ for some $u^* > 0$.

**Assumption C.2** (Distributional Assumptions). We assume the joint distribution $\pi$ of $(X, Y)$ on $\mathbb{R}^d \times \mathbb{R}$ satisfies:

1. $X$ has compact support and a density $\pi_X$ that is bounded away from 0.

2. $|Y - \mathbb{E}[Y \mid X]|$ is almost surely bounded.

3. A version of $\mathbb{E}[Y \mid X]$ is bounded and continuous on $\mathrm{supp}(\pi_X)$.

In some cases above, we have stated slightly stronger forms of the assumptions in Devroye (1978) for brevity, and also because our application is relatively simple and works under these stronger assumptions.

**Theorem C.3** (Theorem 2 in Devroye (1978), Paraphrased). *Let $K$ be a kernel satisfying Assumption C.1 with a sequence of bandwidths $\sigma_n$, and suppose $\pi$ satisfies Assumption C.2. Then if $\sigma_n \to 0$ and $n\sigma_n^{2d}/\log n \to \infty$, we have uniform convergence of the kernel regression estimator $\hat{m}_{\sigma_n}$:*

$$\underset{x \in \mathrm{supp}(\pi_X)}{\mathrm{ess\,sup}} |\hat{m}_{\sigma_n}(x) - \mathbb{E}[Y \mid X = x]| \to 0 \text{ a.s.} \tag{C.2}$$

We now state and prove the formal version of Theorem 4.9.

**Theorem C.4.** *Let $\hat{m}_\sigma$ be as in (4.8). Given a sequence of bandwidths $\sigma_n$, we define the plugin estimator $\hat{d}_{\varphi,n}$ as follows:*

$$\hat{d}_{\varphi,n} = \frac{1}{n} \sum_{i=1}^n \varphi\big(\hat{m}_{\sigma_n}(h(x_i))\big) - \varphi\big(h(x_i)\big) - \varphi'\big(h(x_i)\big)\big(\mathbb{1}_{y_i = c(x_i)} - h(x_i)\big). \tag{C.3}$$

*Suppose $K$ satisfies Assumption C.1 and $\sigma_n$ satisfies $\sigma_n \to 0$ and $n\sigma_n^2/\log n \to \infty$. If we additionally have that $E[\mathbb{1}_{Y=c(X)} \mid h(X) = h(x)]$ is continuous in $h(x)$ and that $h(X)$ has a density bounded away from zero, it follows that:*

$$\hat{d}_{\varphi,n} \to \mathbb{E}[d_\varphi\big(\mathbb{E}[\mathbb{1}_{Y=c(X)} \mid h(X)], h(X)\big)] \tag{C.4}$$

*in probability.*

*Proof.* By definition and properties of conditional expectation (covariance matching, the same trick we used in the proof of Lemma 4.1), we have that:

$$\mathbb{E}\left[d_\varphi\big(\mathbb{E}[\mathbb{1}_{Y=c(X)} \mid h(X)], h(X)\big)\right] = \mathbb{E}[\varphi\big(\mathbb{E}[\mathbb{1}_{Y=c(X)} \mid h(X)]\big)] - \mathbb{E}[\varphi\big(h(X)\big)]$$
$$- \mathbb{E}[\varphi'(h(X))(\mathbb{1}_{Y=c(X)} - h(X))]. \tag{C.5}$$

It is immediate from the law of large numbers that the second and third terms in (C.3) converge to their counterparts in (C.5). It remains to show the convergence of the first term of (C.3).

We observe that since $\mathbb{1}_{Y=c(X)} \leq 1$, along with the fact that we have assumed $\mathbb{E}[\mathbb{1}_{Y=c(X)} \mid h(X) = h(x)]$ is continuous in $h(x)$ and that $h(X)$ has a density bounded away from zero, we satisfy all of the conditions of Assumption C.2. Thus we have uniform convergence of $\hat{m}_{\sigma_n}$ from Theorem C.3.

It follows from uniform convergence and the boundedness of $\mathbb{E}[\mathbb{1}_{Y=c(X)} \mid h(X)]$ that $\hat{m}_{\sigma_n}(h(X))$ is constrained to a compact set for $n$ sufficiently large. Since $\varphi$ is uniformly continuous on a compact set, we then have that:

$$\mathbb{E}\left[\frac{1}{n}\sum_{i=1}^{n}\varphi\big(\hat{m}_{\sigma_n}(h(x_i))\big)\right] = \mathbb{E}[\varphi\big(\mathbb{E}[\mathbb{1}_{Y=c(X)} \mid h(X)]\big)] + \delta_n \qquad \text{(C.6)}$$

for some $\delta_n \to 0$ as $n \to \infty$. The result C.4 then immediately follows. □

## D  CALIBRATION-SHARPNESS DIAGRAM DETAILS

Here we go over the fine-grained details of constructing calibration-sharpness diagrams, beginning with estimation of the conditional expectations involved.

**Estimation details.** We estimate the term $\mathbb{E}\left[d_\phi(\pi_{Y|X}, g(X)) \mid h(X)\right]$ in Equation (4.7) by replacing $(x_i, y_i)$ data point pairs with $(x_i, d_\phi(e_{y_i}, g(x_i)))$ (i.e. the divergence of $g(x_i)$ from the one-hot encoded label $y_i$) and then using the kernel regression estimator (4.8). The same is done with $\mathbb{E}[\mathbb{1}_{Y=c(X)} \mid h(X)]$, except the label is replaced with pointwise accuracy, i.e. $\mathbb{1}_{y_i=c(x_i)}$. We also estimate the density $p_{h(X)}$ using a kernel density estimate (this is just the denominator of the kernel regression (4.8)) with the same kernel, and refer to this estimate as $\hat{p}_{h(X)}$.

With these estimates, we can estimate the pointwise sharpness gap $\rho(h(x))$. We visualize this gap as a band whose length is $\hat{p}_{h(X)}(h(x))\rho(h(x))$ centered around the calibration curve $\mathbb{E}[\mathbb{1}_{Y=c(x)} \mid h(x)]$. Note that we scale the band by $\hat{p}_{h(X)}(h(x))$, as this is important for making it visually obvious where most of the sharpness error is concentrated and also allows the aggregated total sharpness gap area to better correspond to the actual total sharpness gap.

**Visualization details.** Our kernel regression estimates for the diagrams in Figure 1 were computed using a Gaussian kernel with $\sigma = 0.05$ (different choices of kernel and bandwidth are discussed in Appendix H). Due to the size of ImageNet-1K Val, we also replace the full kernel regression estimate of (4.8) with a mean over 10 estimates in which we randomly subsample the data to 5000 data points, so as to fit the entire computation into memory. The standard deviations of these estimates is shown alongside the mean calibration error scaled by 100 ($d_{\varphi,\text{CAL}}$ in all of the plots); in all cases they are negligible, and we checked that subsampling loses virtually nothing. Lastly, for stylistic reasons, we opt to threshold the curve defining the bottom of the sharpness band at 0, but do not threshold the top curve at 1. In Figure 1 and the plots of Appendix H, we do not align the Y-axes of the plots due to the large differences in sharpness gaps between some methods. However, for direct comparisons between two approaches (or a handful of similar approaches) we recommend aligning the Y-axes as we have done in Figure 2.

**A note on estimating conditional expectations.** We have used the Nadaraya-Watson estimator for the conditional expectation terms in all of the above as it is the standard choice for nonparametric estimation and is known to be minimax optimal under certain technical conditions on the data distribution and the choice of kernel (Tsybakov, 2008). However, if there is some prior knowledge of the parametric form of $\mathbb{E}[Y \mid g(X)]$, then one can of course achieve better estimation results using such a form, but we expect such forms would not be known in most applications. For further details regarding the asymptotic properties (bias, consistency, convergence rates) of Nadaraya-Watson style estimators we recommend Tsybakov (2008) and Györfi et al. (2002).

## E  DETAILED COMPARISON TO PRIOR WORK ON GROUPING LOSS

We have focused on using the sharpness-calibration decomposition of Bregman divergences to better understand model performance, but there exist other decompositions in the proper scoring rule literature that can provide different perspectives on performance. In particular, Kull & Flach (2015) proposed a decomposition of divergences derived from proper scoring rules that splits the total divergence into calibration error (like we have in Lemma 4.1) as well as *grouping loss* and *irreducible loss*.

To make these notions precise, let us for simplicity consider a binary classification problem with $Y \in \{0,1\}, X \in \mathbb{R}^d$ and some classifier $g : \mathbb{R}^d \to [0,1]$. In this context, we recall that the calibration condition is simply $\mathbb{E}[Y \mid g(X)] = g(X)$. Now let us further define $C = \mathbb{E}[Y \mid g(X)]$ and $Q = \mathbb{E}[Y \mid X]$. Kull & Flach (2015) showed that, for any divergence $d_\phi$ derived from a proper scoring rule (to keep background discussion brief, one can just think of this as one of the Bregman divergences we consider in the main paper), we have:

$$\mathbb{E}[d_\phi(Y, g(X))] = \underbrace{\mathbb{E}[d_\phi(Y, Q)]}_{\text{Irreducible Loss}} + \underbrace{\mathbb{E}[d_\phi(Q, C)]}_{\text{Grouping Loss}} + \underbrace{\mathbb{E}[d_\phi(C, g(X))]}_{\text{Calibration Error}}. \qquad \text{(E.1)}$$

Here the irreducible loss captures the uncertainty inherent to the problem as it does not depend on the classifier $g$, while the grouping loss captures the deviation of the conditional expectation using the predicted confidences from the true conditional expectation derived from the underlying data. Intuitively, the grouping loss measures how much information from the data is "grouped" together by our classifier – for example, if $g$ is a bijection, then this term is zero since $g(X)$ generates the same $\sigma$-algebra as $X$. Our interpretation of sharpness as "resolution" or "fine-grainedness" of the model can be attributed to this grouping loss term, as also discussed in Pohle (2020).

The key idea in the work of Perez-Lebel et al. (2023) is that this grouping loss term should be reported alongside calibration error and other metrics, and they provide similar examples to what we provide in the main paper for how a model can be calibrated and have perfect accuracy but not recover the ground-truth distribution. The main results of Perez-Lebel et al. (2023) consequently regard the estimation of grouping loss as well as joint visualizations of the grouping loss and calibration error.

In the multi-class setting, the results of Perez-Lebel et al. (2023) are applied directly to the confidence calibration problem. However, our results in Section 4 tie together the full multi-class problem with the confidence calibration problem; this allows us to not reduce the entire problem to confidence calibration (where the labels are binary and the probabilities are 1-D). This is important because ultimately we care about our generalization metrics in the context of the original multi-class problem, and the purpose of Lemma 4.6 was to show that we can simply replace the full, high-dimensional calibration error with its 1-D confidence calibration counterpart while still maintaining a reasonable interpretation.

Additionally, by working with the sharpness gap as opposed to grouping loss and irreducible loss, we need only concern ourselves with the estimating the conditional expectation $\mathbb{E}[Y \mid g(X)]$ as opposed to also considering $\mathbb{E}[Y \mid X]$. The inclusion of the irreducible loss in the sharpness gap is not an issue; as this is shared by all models, it does not have an impact on how we rank them relative to one another. To summarize, our approach differs in that we tie the calibration-sharpness decomposition in the full calibration setting to confidence calibration as opposed to working with grouping loss in the confidence calibration setting, which leads to easier estimation and maintains important properties of the original multi-class problem.

## F   ADDITIONAL EXPERIMENTS WITH ALTERNATIVE MODELS

### F.1   FURTHER IMAGENET EXPERIMENTS

We focused on recalibrating a pretrained ViT model on ImageNet throughout the main paper for consistency across sections; here we consider different model backbones applied to ImageNet and recreate the results of Table 1 and Figure 1 for each of the different models. As with the ViT model, we use pretrained models from the `timm` library that have strong performance on ImageNet. In particular, we consider ResNet-50 (He et al., 2015), EfficientNet (Tan & Le, 2019), and ConvNext (Liu et al., 2022) backbones. Final tabular results are shown in Tables 2 to 4, and corresponding plots are shown in Figures 3 to 5.

### F.2   CIFAR EXPERIMENTS

We used ImageNet in the main paper due to the plethora of high quality, pretrained models available via open source libraries. To show that our observations extend to other datasets, however, we also include experiments on CIFAR-10 and CIFAR-100. In particular, we use the pretrained ResNet-32 models of Yaofo Chen (`https://github.com/chenyaofo/pytorch-cifar-models`) and calibrate them using 20% of the CIFAR-10 and CIFAR-100 test sets to be consistent with our ImageNet experiments. Numerical results are shown in Tables 5 and 6, and plots are shown in Figures

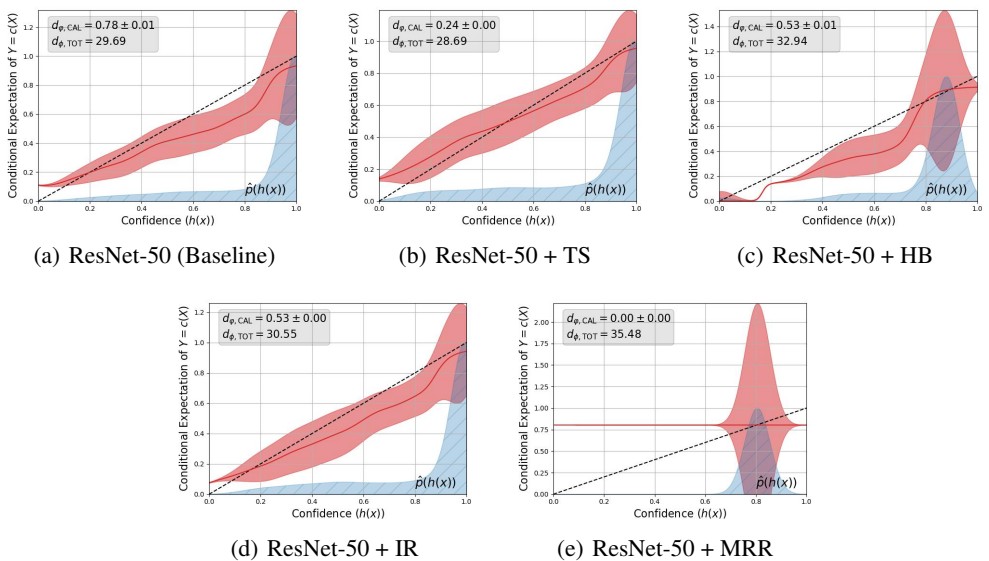

Figure 3: Calibration-sharpness diagrams using MSE/Brier score for the ResNet-50 experiments of Table 2.

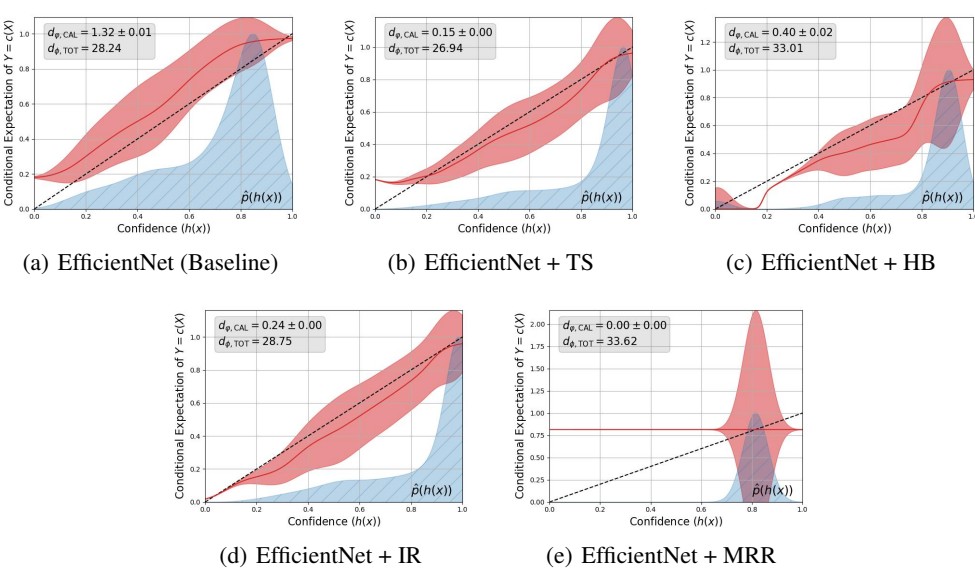

Figure 4: Calibration-sharpness diagrams using MSE/Brier score for the EfficientNet experiments of Table 3.

| Model | Test Accuracy | ECE | ACE | SmoothECE | NLL | MSE/Brier Score |
|---|---|---|---|---|---|---|
| ResNet-50 (Baseline) | **80.33** | 8.68 | 10.48 | 8.26 | 93.98 | 29.69 |
| ResNet-50 + TS | **80.33** | 4.97 | 6.00 | 4.85 | **88.63** | **28.69** |
| ResNet-50 + HB | 78.70 | 6.44 | 15.11 | 4.77 | ∞ | 32.94 |
| ResNet-50 + IR | 79.12 | 7.25 | 8.81 | 7.20 | ∞ | 30.55 |
| ResNet-50 + MRR | **80.33** | **0.28** | **0.28** | **0.28** | 185.48 | 35.48 |

Table 2: Table 1 in the main paper recreated using a pretrained ResNet-50 backbone (timm/resnet50.a1_in1k).

| Model | Test Accuracy | ECE | ACE | SmoothECE | NLL | MSE/Brier Score |
|---|---|---|---|---|---|---|
| EfficientNet (Baseline) | **81.47** | 10.66 | 9.97 | 10.67 | 85.50 | 28.24 |
| EfficientNet + TS | **81.47** | 3.44 | 5.85 | 3.41 | **75.17** | **26.94** |
| EfficientNet + HB | 77.63 | 5.44 | 12.92 | 4.58 | ∞ | 33.01 |
| EfficientNet + IR | 80.25 | 4.92 | 7.80 | 4.88 | ∞ | 28.75 |
| EfficientNet + MRR | **81.47** | **0.07** | **0.07** | **0.07** | 175.92 | 33.62 |

Table 3: Table 1 in the main paper recreated using a pretrained EfficientNet backbone (timm/efficientnet_b3.ra2_in1k).

| Model | Test Accuracy | ECE | ACE | SmoothECE | NLL | MSE/Brier Score |
|---|---|---|---|---|---|---|
| ConvNeXt (Baseline) | **84.13** | 6.93 | 6.80 | 6.94 | 68.34 | 23.78 |
| ConvNeXt + TS | **84.13** | 3.00 | 4.70 | 2.99 | **62.01** | **23.38** |
| ConvNeXt + HB | 80.96 | 3.64 | 12.60 | 4.09 | ∞ | 28.68 |
| ConvNeXt + IR | 82.72 | 4.93 | 8.29 | 4.84 | ∞ | 25.57 |
| ConvNeXt + MRR | **84.13** | **0.13** | **0.13** | **0.13** | 153.34 | 29.22 |

Table 4: Table 1 in the main paper recreated using a pretrained ConvNeXt backbone (timm/convnext_tiny.in12k_ft_in1k).

6 and 7. The comparison between the CIFAR-10 and CIFAR-100 results highlights the advantage of the sharpness gap bands in the calibration-sharpness diagrams: we see that while calibration does not suffer too much between CIFAR-10 and CIFAR-100, sharpness is much worse on CIFAR-100 and consequently generalization metrics (MSE/Brier Score) are also much worse.

| Model | Test Accuracy | ECE | ACE | SmoothECE | NLL | MSE/Brier Score |
|---|---|---|---|---|---|---|
| ResNet-32 (Baseline) | **93.46** | 4.15 | 16.01 | 3.94 | 29.21 | 10.87 |
| ResNet-32 + TS | **93.46** | 0.72 | 5.79 | 0.88 | **21.86** | **10.00** |
| ResNet-32 + HB | 93.12 | 1.45 | 15.17 | 1.46 | ∞ | 11.75 |
| ResNet-32 + IR | 93.29 | 1.33 | 5.11 | 1.34 | ∞ | 10.18 |
| ResNet-32 + MRR | **93.46** | **0.34** | **0.34** | **0.34** | 38.52 | 12.60 |

Table 5: Pretrained ResNet-32 model + recalibration approaches evaluated on CIFAR-10.

| Model | Test Accuracy | ECE | ACE | SmoothECE | NLL | MSE/Brier Score |
|---|---|---|---|---|---|---|
| ResNet-32 (Baseline) | **70.20** | 13.25 | 15.71 | 13.12 | 132.31 | 44.08 |
| ResNet-32 + TS | **70.20** | 1.71 | 2.42 | 1.53 | **111.37** | **40.98** |
| ResNet-32 + HB | 65.76 | 9.56 | 18.10 | 6.97 | ∞ | 49.84 |
| ResNet-32 + IR | 68.70 | 6.81 | 7.87 | 6.80 | ∞ | 43.02 |
| ResNet-32 + MRR | **70.20** | **0.20** | **0.20** | **0.20** | 197.85 | 50.63 |

Table 6: Pretrained ResNet-32 model + recalibration approaches evaluated on CIFAR-100.

# G    COMPARISON TO SMOOTHECE

As we mentioned in Section 5, the SmoothECE of Błasiok & Nakkiran (2023) is an alternative to ECE that is calculated and visualized using a similar kernel regression approach to ours. The differences

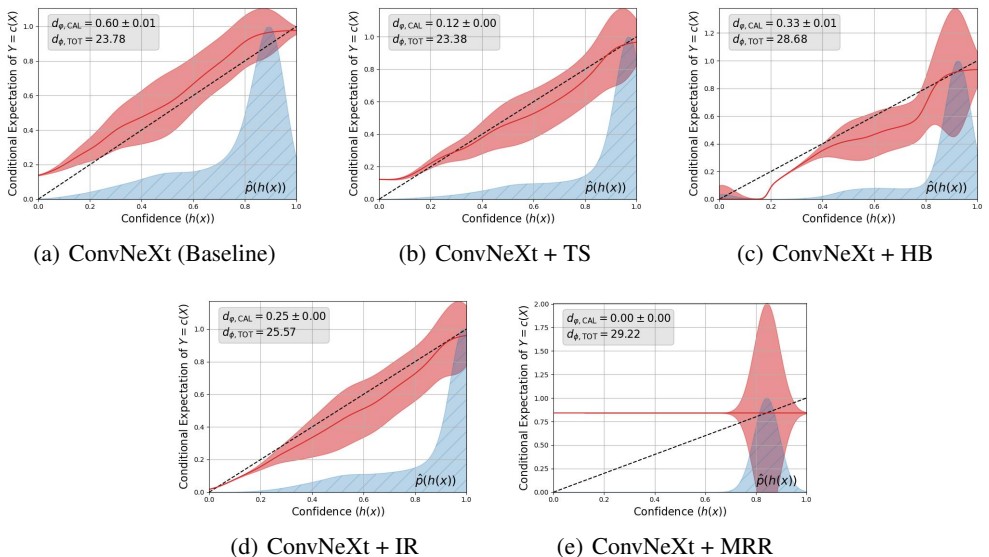

(a) ConvNeXt (Baseline)   (b) ConvNeXt + TS   (c) ConvNeXt + HB

(d) ConvNeXt + IR   (e) ConvNeXt + MRR

Figure 5: Calibration-sharpness diagrams using MSE/Brier score for the ConvNeXt experiments of Table 4.

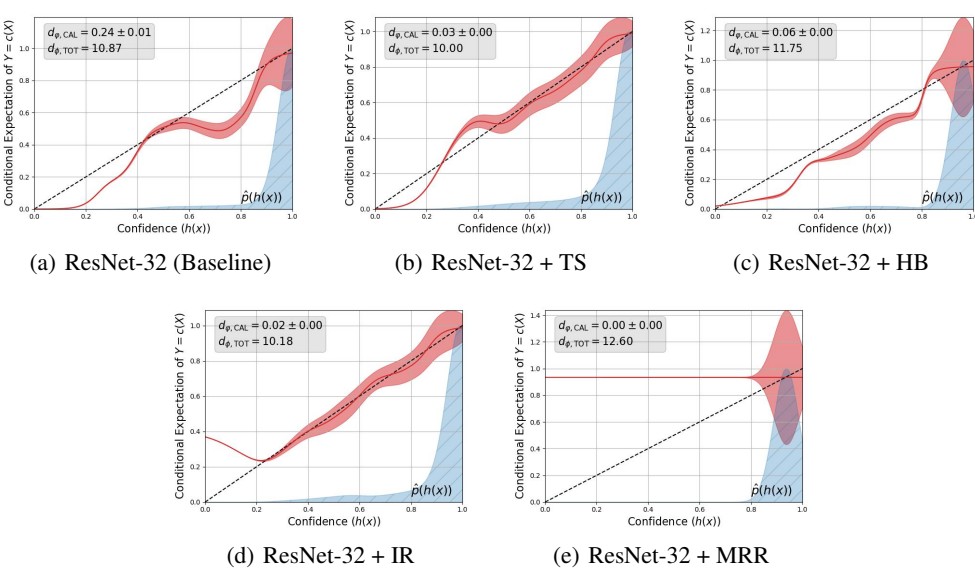

(a) ResNet-32 (Baseline)   (b) ResNet-32 + TS   (c) ResNet-32 + HB

(d) ResNet-32 + IR   (e) ResNet-32 + MRR

Figure 6: Calibration-sharpness diagrams using MSE/Brier score for the ResNet-32 experiments of Table 5 on CIFAR-10.

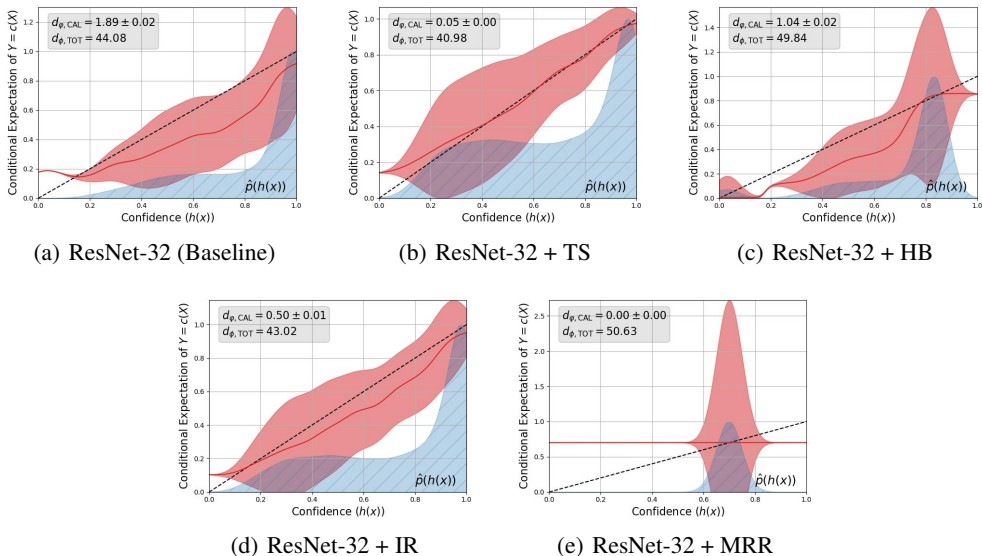

Figure 7: Calibration-sharpness diagrams using MSE/Brier score for the ResNet-32 experiments of Table 6 on CIFAR-100.

between our two approaches are that Błasiok & Nakkiran (2023) use a particular choice of kernel and kernel bandwidth motivated by the framework of Błasiok et al. (2023), and they do not include any kind of notion of sharpness in their visualization. Additionally, they show the density estimate of the confidences using the thickness of the calibration curve as opposed to an overlain density plot.

We show a SmoothECE analogue to Figure 1 in Figure 8. Due to memory constraints, we plot the visualization after subsampling the test data to 5000 data points. While the SmoothECE plots still pick up issues with the MRR recalibration strategy, they make it much trickier to compare histogram binning (HB) and isotonic regression (IR) as we did in Section 5. As we mentioned, for these cases the visualization of sharpness is crucial for understanding the tradeoffs between the methods.

## H  EFFECT OF KERNEL HYPERPARAMETERS

Here we examine the importance of the hyperparameters (choice of kernel and kernel bandwidth) in the calibration-sharpness diagrams of Section 5. The main takeaway from this section is that, for a range of different (reasonable) kernel choices and bandwidths results remain consistent.

**A note on choosing kernel hyperparameters in practice.** Our choice of using a bandwidth of $0.05$ in the main paper was motivated by the recent Logit-Smoothed ECE of Chidambaram et al. (2024), where taking the bandwidth to be the inverse of the number of bins used in binning estimators worked as a good analogue (and 15 to 20 bins is standard practice for datasets of the size we consider). However, in "real life" one can of course tune both the choice of kernel and the kernel bandwidth using held-out data. In particular, we may have a set of models and various estimated (or known) properties of these models (e.g. model A has poor performance whenever its confidences are in the range $[0.7, 0.8]$), and we can tune our kernel/bandwidth to best distinguish these properties across models. This is essentially also how we qualitatively evaluate the different figures in the following two subsections, since our goal is to produce visualizations that display the results of Table 1 in as precise a manner as possible.

### H.1  THE IMPACT OF KERNEL BANDWIDTH

We first consider the same Gaussian kernel used in Section 5 but vary the bandwidth between $0.01$ and $0.25$. We find that values that are less than $0.01$ lead to significant under-smoothing, and values that are larger than $0.1$ lead to over-smoothing. Values in the range $[0.01, 0.05]$ appear to work best. We visualize the choices of $0.01, 0.03, 0.1$, and $0.25$ in Figures 9 to 12 respectively.

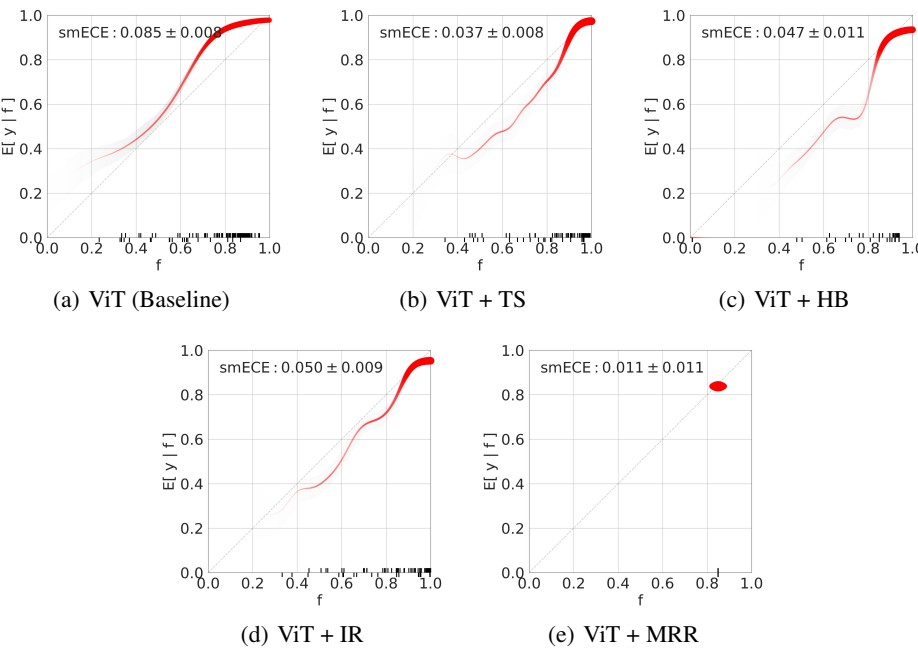

Figure 8: SmoothECE visualization analogue to Figure 1.

We note that when we over-smooth, the ranking of calibration error can change across methods (as shown in Figures 11 and 12). This is similar to known issues with reporting binned ECE in which only a small number of bins are used.

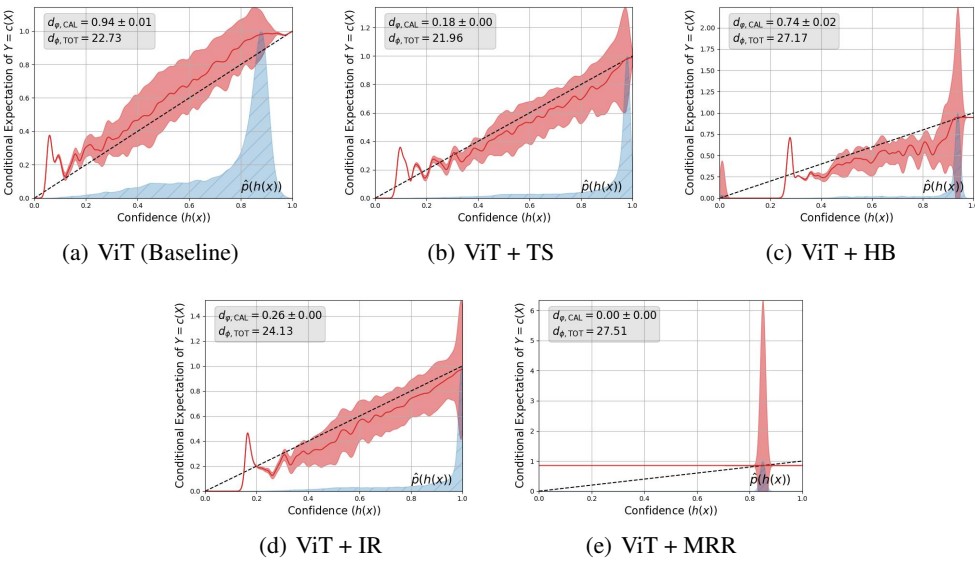

Figure 9: The experiments of Figure 1 but using a bandwidth parameter of 0.01.

## H.2   THE IMPACT OF KERNEL CHOICE

We also consider the impact of changing the kernel used for the visualizations of Figure 1. In particular, we consider the popular choice of the Epanechnikov kernel, which is $K(u) = (3(1 - u^2)/4)\mathbb{1}_{|u| \leq 1}$ in the 1D setting. Results using this kernel with a bandwidth of 0.03 and 0.05 are shown in Figures 13 and 14.

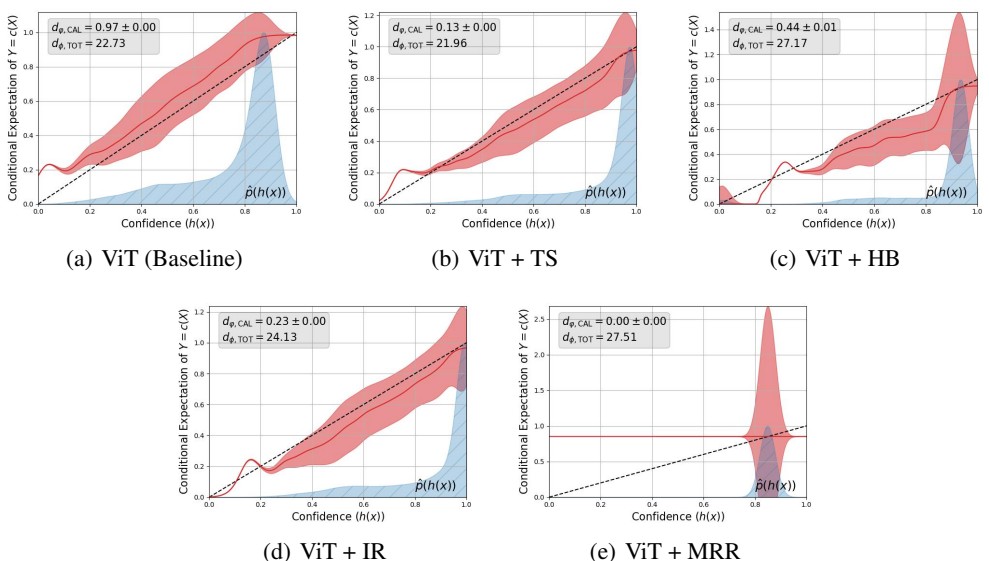

Figure 10: The experiments of Figure 1 but using a bandwidth parameter of 0.03.

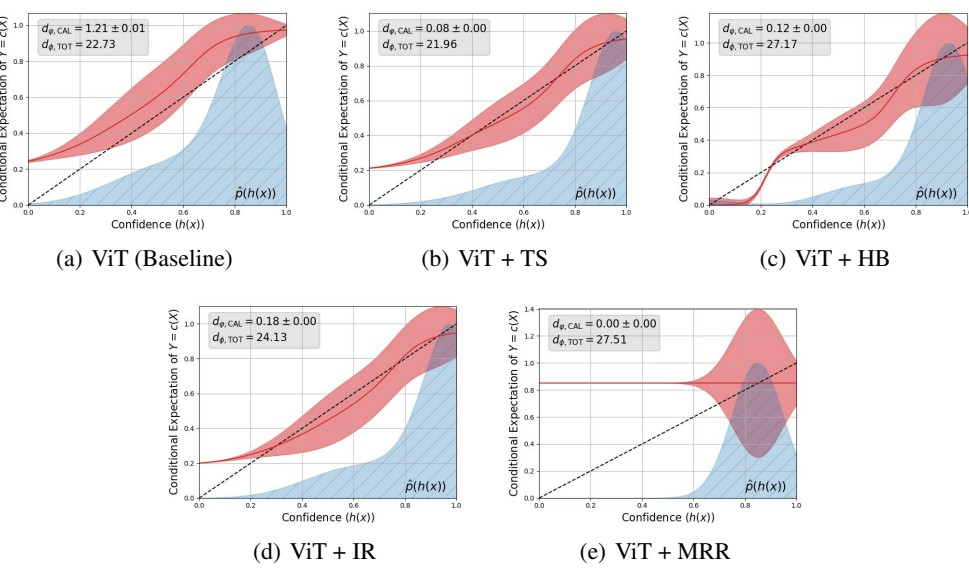

Figure 11: The experiments of Figure 1 but using a bandwidth parameter of 0.1.

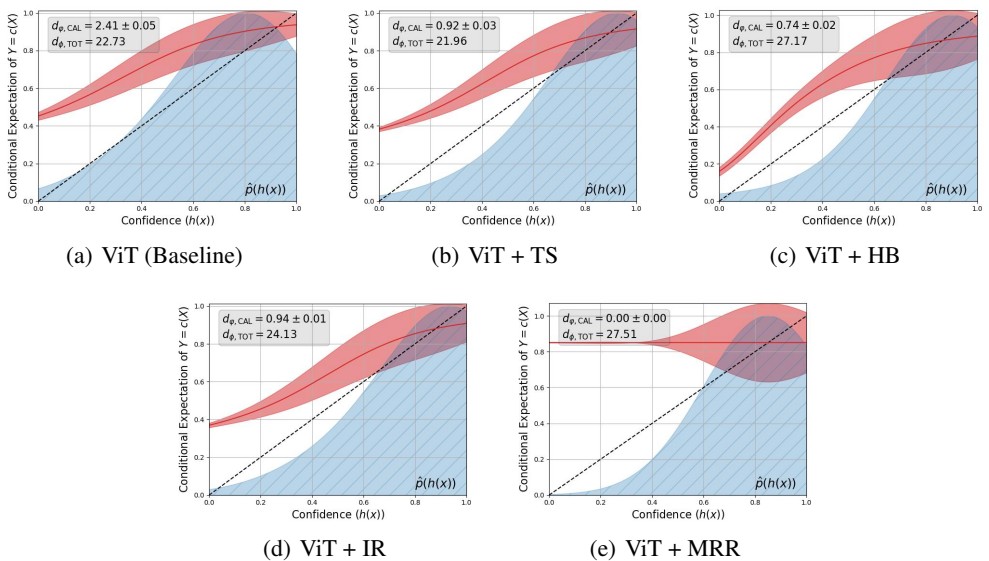

Figure 12: The experiments of Figure 1 but using a bandwidth parameter of 0.25.

As can be seen from the results, changing the kernel does not change the ranking of the recalibration methods, although it does impact the actual reported calibration error. We find that the range of kernel bandwidths suitable in the Gaussian case also applies here.

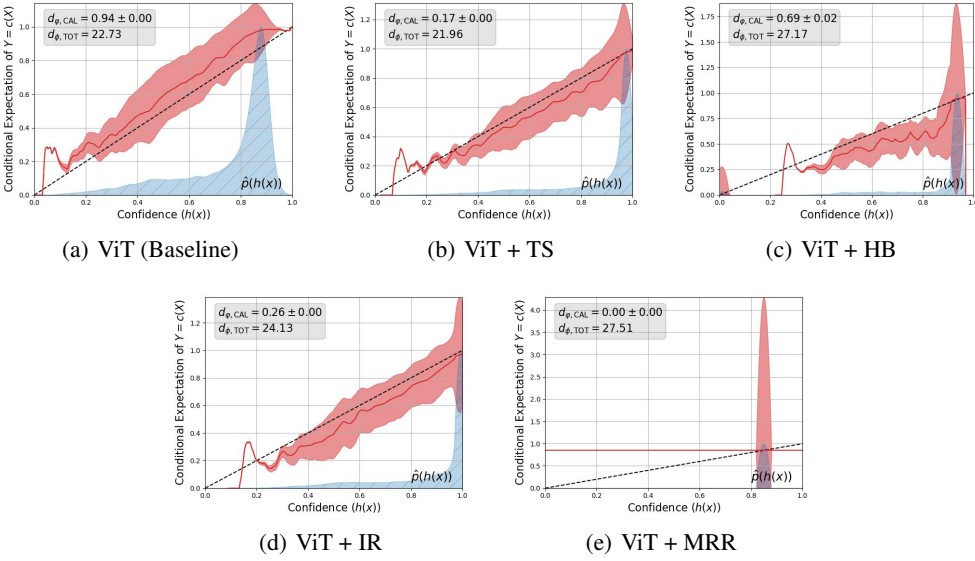

Figure 13: Epanechnikov kernel using a bandwidth parameter of 0.03.

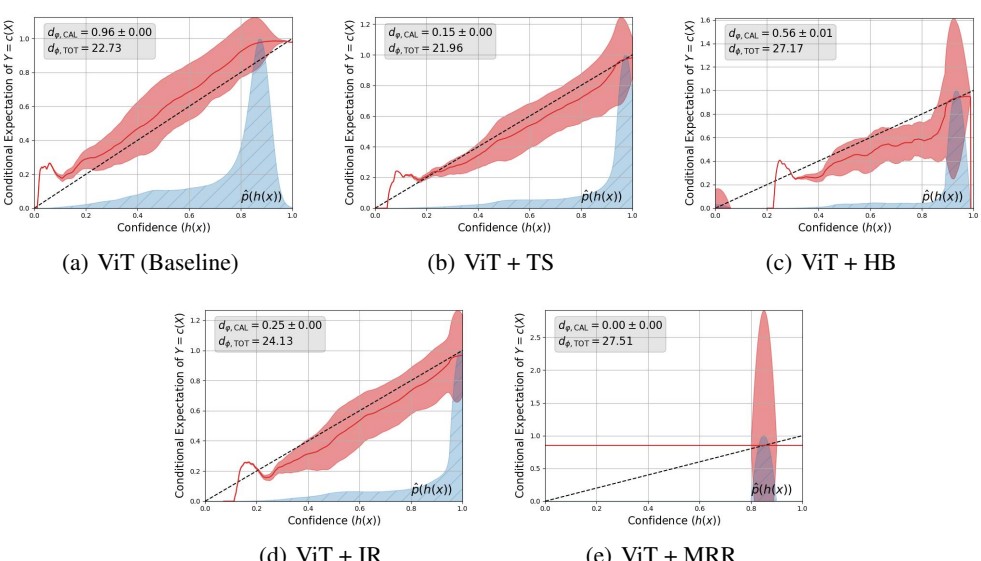

Figure 14: Epanechnikov kernel using a bandwidth parameter of 0.05.

