# OpenReview forum: "Reassessing How to Compare and Improve the Calibration of Machine Learning Models"
_ICLR.cc/2025/Conference — ICLR 2025 Poster_

### Official Review · Reviewer_Dhyc · 2024-10-31

**Soundness:** 3
**Presentation:** 2
**Contribution:** 2
**Rating:** 5
**Confidence:** 3

**Summary:**

This paper studies the task of calibration: measuring how well a model predicts the class probability and improving the quality.
It begins with studying the Expected Calibration Error (ECE), a popular metric of how well a model predicts the class probability.
The authors point out that the ECE can be improved by a trivial modification to the confidence function. The authors present experiments (Table 1) showing how this metric differs from proper scoring rules such as the Mean Squared Error (MSE) and the Negative Log Loss (NLL).

The paper then moves on to the study about proper scoring rules induced by Bregman divergences. The authors explain the gap between a proper scoring rule and the calibration error using a decomposition of Bregman divergences (Lemma 4.1) with some properties of this gap (Proposition 4.2 and 4.3). The pointwise variant of the gap is also defined (Eq. (4.7)).

The authors also show that the gap between the full calibration error and its one-dimensional counterpart.

Finally, the paper presents visualization results based on the proposed metric (4.7).

**Strengths:**

- Theorems and propositions are interesting and seem correct. They are supported by rigorous proofs.
- The paper studies an important problem of calibration.
- The paper provides a good literature review, which motivates the direction of this work well.

**Weaknesses:**

- The paper explains ECE can be improved by a trivial modification to the confidence function but does not explain why it is a bad thing. If estimating the whole class probabilities is not the objective and we only care about the confidence of the predicted class, this solution could be just fine.

- The descriptions of the method and the results are not clear to me.

- The benefits of the visualization are not clear from the results. In particular, I do not agree with the claim that "looking at the calibration-sharpness diagrams the differences between the two approaches becomes more clear."

- The notation in some parts is a little confusing.

- Some of the proofs lack details, and there are also some writing mistakes. These make the proofs a little difficult to follow.

**Questions:**

### Major comments
- Are some of the findings of this paper related to those of Prez-Lebel et al. (2023)?

Alexandre Perez-Lebel, Marine Le Morvan, and Gaël Varoquaux. BEYOND CALIBRATION: ESTIMATING THE GROUPING LOSS OF MODERN NEURAL NETWORKS. ICLR, 2023.

- Corollary 4.7 should be more formally stated.

- I could not understand the condition $d_{\phi}(x, y) \ge d_{\psi}(x_i, y_i)$.

- Is there any justification for thinking about the band proposed in lines 419-425? Also, the description here is not very clear to me despite its importance.

- There should be a part explaining how to read the plots. What are the read curves, red areas, and blue areas? Then, there should be explanations about how to evaluate them such as what kind of curves and areas are supposed to be good. Some explanations of them are mixed with the interpretations of the results, which is making the section difficult to read.

### Minor comments
- The notation in Eq. (2.1) feels strange. If $\mathbb{P}( \cdot | \cdot )$ denotes the regular conditional distribution, it should take an event in the first argument. It is also strange that it returns a vector, assuming $v \in \mathbb{R}^m$. Could the authors check the definition again?

- I think the sign is not correct in (A.1).

- The part of the proof obtaining (B.2) should be explained more clearly.

- The proof of Corollary C.1 could be more precise.

- Could the authors add a clear description about the standard reliability diagram?

- I suggest the authors explicitly explain all the acronyms in the paper.

---

> ### Author Response · Authors · 2024-11-15
> **Response to Reviewer Dhyc (Part 1)**
>
> We would like to thank Reviewer Dhyc for taking the time to review our paper, and we appreciate the positive comments regarding our contributions as well the precise suggestions for revisions. We hope to address all of the expressed concerns below.
>
> ## Weaknesses
> > The paper explains ECE can be improved by a trivial modification to the confidence function but does not explain why it is a bad thing. If estimating the whole class probabilities is not the objective and we only care about the confidence of the predicted class, this solution could be just fine.
>
> We do not disagree that if we only care about having good (aggregate) confidence calibration and accuracy, there is nothing fundamentally wrong with mean replacement recalibration, i.e. just always using our aggregate confidence. However, the point we wanted to make was that typically people want more than that -- if a model always predicts a constant (or near-constant) confidence, this is likely not useful for downstream decision-makers who would want to make different decisions at different confidence levels.
>
> > The descriptions of the method and the results are not clear to me.
>
> In the revision, we have added an explicit discussion in Section 4.2 of how to interpret the components of our plots, which we hope makes the results more clear.
>
> > The benefits of the visualization are not clear from the results. In particular, I do not agree with the claim that "looking at the calibration-sharpness diagrams the differences between the two approaches becomes more clear."
>
> We apologize -- we realize now that the example we chose to illustrate the differences between reliability diagrams and our approach is perhaps not the most obvious. We have replaced the comparison in Section 5.2 with a comparison between Histogram Binning and Isotonic Regression, where we think it should be more clear what is to be gained from using the calibration-sharpness diagrams.
>
> > The notation in some parts is a little confusing.
>
> We have added some more text to explain notation in parts where it may have been confusing, i.e. Section 4.1.
>
> > Some of the proofs lack details, and there are also some writing mistakes. These make the proofs a little difficult to follow.
>
> We have added more detail to all proofs that were pointed out to be too brief, and also fixed typos (thank you for catching them).
>
> ## Major Comments
>
> > Are some of the findings of this paper related to those of Prez-Lebel et al. (2023)?
>
> Thank you for mentioning the work of Perez-Lebel et al. (2023), it is indeed very relevant to our work and we sincerely apologize not discussing it appropriately in the submitted version of the paper. Perez-Lebel et al. (2023) puts forth a similar message to us, but we achieve this using two different approaches (analyzing grouping loss vs. analyzing sharpness). The grouping loss results have to be applied to the confidence calibration problem directly (i.e. reduce to the binary setting where labels are 0, 1 depending on whether we predicted correctly or not), but our results allow us to connect the full generalization error in the multi-class setting with the 1-D confidence calibration error as we discuss in Section 4. We have now referenced Perez-Lebel et al. (2023) in the paper, and have included a more comprehensive discussion in Appendix E comparing our two works.
>
> > Corollary 4.7 should be more formally stated.
>
> Corollary 4.7 is now stated fully formally, and the proof also has much more detail than before.
>
> > I could not understand the condition $d_{\phi}(x, y) \ge d_{\varphi}(x_i, y_i)$.
>
> Here we recall that $d_{\phi}$ is defined on $\mathbb{R}^k \times \mathbb{R}^k$ whereas $d_{\varphi}$ is defined on $\mathbb{R} \times \mathbb{R}$; i.e. the latter is a "1-D counterpart" to the former. The condition is then that the total divergence $d_{\phi}$ dominates this 1-D divergence applied to any coordinate pair; a simple example is $||x - y||^2$ compared to $(x_i - y_i)^2$ since the former is a sum over the latter for all $i$. We have made this clearer in the discussion preceding Lemma 4.6.

---

> > ### Author Response · Authors · 2024-11-15
> > **Response to Reviewer Dhyc (Part 2)**
> >
> > > Is there any justification for thinking about the band proposed in lines 419-425? Also, the description here is not very clear to me despite its importance.
> >
> > The sharpness gap band $\rho(h(x))$ can be thought of as the error of the model at a confidence level of $h(x)$ that is not attributable to miscalibration, i.e. this is leftover generalization error. In that sense, the band can be thought of as visualizing generalization error conditioned on different confidence levels. Intuitively, sharpness can also be thought of as model "resolution" as we point out in the discussions before Propositions 4.2 and 4.3 -- for example, a model that only ever predicts with a handful of different confidence levels (e.g. a binned predictor) can have poor resolution.
> >
> > > There should be a part explaining how to read the plots. What are the read curves, red areas, and blue areas? Then, there should be explanations about how to evaluate them such as what kind of curves and areas are supposed to be good. Some explanations of them are mixed with the interpretations of the results, which is making the section difficult to read.
> >
> > Thank you for the suggestion of having a separate discussion of how to read the calibration-sharpness plots; we agree this is useful to have and now have such a discussion in Section 4.2, and have opted to relegate more technical details of the visualization to Appendix D.
> >
> > ## Minor Comments
> >
> > > The notation in Eq. (2.1) feels strange. If $\mathbb{P}(\cdot \mid \cdot)$ denotes the regular conditional distribution, it should take an event in the first argument. It is also strange that it returns a vector, assuming $v \in \mathbb{R}^m$. Could the authors check the definition again?
> >
> > We apologize for the slight abuse of notation with respect to the conditional distribution; you are right in that typically one thinks of regular conditional probabilities as Markov transition kernels mapping points in the sample space to measures. Here when we write $\mathbb{P}(Y \mid g(X) = v)$ we are specifying the entire measure; the variable $Y$ can only take one of $k$ values. We discuss this identification of the distribution with a vector in $\mathbb{R}^k$ right above Equation (2.1) in the notation section.
> >
> > > I think the sign is not correct in (A.1).
> >
> > Thank you for catching the typo in Equation (A.1), it is now fixed in the revision.
> >
> > > The part of the proof obtaining (B.2) should be explained more clearly.
> >
> > We have added more detail showing how we arrive at Equation (B.2).
> >
> > > The proof of Corollary C.1 could be more precise.
> >
> > Thank you for the point regarding Corollary (C.1); since the result is actually more general than the originally stated Lemma 4.6 (we obtain the original lemma from just taking expectations), we have updated Lemma 4.6 to display this result and also updated the proof of Lemma 4.6 accordingly.
> >
> > > Could the authors add a clear description about the standard reliability diagram?
> >
> > We have now included a description of the standard reliability diagram in the experiments section where we compare to it. Furthermore, we have also included new comparisons to the diagrams produced by SmoothECE [1] as well in Appendix G.
> >
> > > I suggest the authors explicitly explain all the acronyms in the paper.
> >
> > We have combed through the paper to make sure all acronyms are defined prior to their use; please let us know if you see any that aren't and we will be happy to fix them.
> >
> > We hope the above revisions address the reviewer's main concerns regarding work, and we are happy to answer any further questions that my persist.
> >
> > [1] Jaroslaw Blasiok and Preetum Nakkiran. Smooth ece: Principled reliability diagrams via kernel smoothing, 2023.

---

> > > ### Comment · Reviewer_Dhyc · 2024-11-24
> > >
> > > Dear Authors,
> > >
> > > Is it possible to have a version with the changes highlighted with some color? It's difficult to see what changes have been made.

---

> > > > ### Author Response · Authors · 2024-11-25
> > > >
> > > > Our apologies for the delay -- we have uploaded a version with the main changes highlighted in brick red. Feel free to let us know if you have any further questions or concerns.

---

> > > > > ### Author Response · Authors · 2024-12-01
> > > > >
> > > > > Hi, we just wanted to bump this since the discussion period is ending soon -- is there anything further we could do to answer your questions? We believe the (now highlighted) changes in the revision address the main concerns expressed in your initial review, but we are happy to provide any further clarifications.

---

### Official Review · Reviewer_sELH · 2024-11-02

**Soundness:** 3
**Presentation:** 2
**Contribution:** 3
**Rating:** 6
**Confidence:** 3

**Summary:**

- The authors identify key issues in current calibration performance reporting in machine learning literature, emphasizing that focusing solely on calibration error (e.g., ECE) and accuracy can lead to misleading conclusions.
- They show that simple recalibration strategies may outperform commonly used methods if calibration metrics are not presented alongside generalization metrics, such as negative log-likelihood (NLL).
- The authors develop a theoretical framework that aligns calibration metrics with generalization metrics using Bregman divergences, proposing that calibration measures should be selected based on the chosen generalization metric. They establish novel connections between full calibration error and confidence calibration error.
- They introduce “calibration-sharpness diagrams” as an extension to traditional reliability diagrams, allowing for a joint visualization of calibration and generalization errors and providing a more detailed view of the trade-offs between calibration and model sharpness.

**Strengths:**

- The paper systematically identifies critical issues in current reporting practices for calibration and prediction accuracy, highlighting the limitations of relying solely on metrics like ECE and accuracy.
- By utilizing Bregman divergences, the authors develop a unified framework that aligns calibration metrics with generalization metrics. This theoretical approach offers valuable insights for the development and evaluation of new calibration methods.

**Weaknesses:**

### Concerns for Unclear Guidelines for Selecting Calibration Metrics:
- While the paper proposes selecting calibration metrics based on generalization metrics, it does not provide clear guidelines or step-by-step procedures for applying this principle in practice. From a practical perspective, providing concrete examples and a well-defined workflow for implementing the proposed framework would enhance the paper’s contribution and make its practical applications more accessible.

### Sensitivity of Calibration-Sharpness Diagrams to Kernel and Hyperparameter Selection:
- In calibration-sharpness diagrams based on kernel regression estimators, the choice of kernel type and hyperparameter values appears to significantly influence the results. This concern is relevant not only to the proposed method but also to other calibration error evaluations based on Nadaraya-Watson estimators, such as smooth ECE. However, these parameters pose a risk of being adjusted arbitrarily to produce favorable calibration performance. A more in-depth discussion on minimizing this arbitrariness would help ensure fair calibration evaluation.
- Additionally, the Appendix reveals considerable variability in calibration performance across different kernel types and hyperparameter settings, raising concerns about the reliability of calibration-sharpness diagrams produced by the proposed framework. If calibration performance assessments based on these diagrams are highly sensitive to such choices, further discussion on criteria for kernel selection and strategies for tuning hyperparameters would strengthen the paper’s contributions.

### Concerns for Readability and Presentation Clarity:
- The presentation of certain theoretical elements in the paper could benefit from added clarity, as the connection between some propositions and the main text is not always apparent. For example, in Proposition 4.5, a more detailed explanation of the “always wrong predictor” and its implications could improve understanding. This may stem from a lack of detailed exposition rather than the inherent complexity of the material, but additional clarification would enhance accessibility.
### Concerns for Relatively Weak Explanation for the Motivation Behind Using Bregman Divergences:
- While the paper establishes useful calibration metrics based on Bregman divergences, it does not clearly explain why the authors hypothesized that Bregman divergence-based metrics would be effective for calibration. Specifically, the rationale behind selecting Bregman divergences over other possible metrics remains unclear, making it challenging to fully understand the motivation for the proposed framework. Although the benefits of this choice are evident in the theoretical results, a more explicit discussion on why Bregman divergences were chosen would provide valuable context.
- I understand that the authors’ main motivation for adopting Bregman divergences could be that they provide a general metric encompassing several proper scoring rules, allowing for a unified approach to selecting calibration evaluation metrics. Emphasizing this reasoning could facilitate a smoother transition from Section 3 to Section 4. While this is a personal suggestion and not strictly necessary, including a brief explanation in Section 2 on the basic properties of Bregman divergences and their relationship to calibration metrics might improve the clarity and flow of the presentation.

**Questions:**

Based on the above Weakness points, I would like to ask the following questions. If appropriate and sufficient discussion is provided on these points, I would be pleased to consider raising my score.

- Could you provide specific examples or a more detailed workflow for selecting calibration metrics based on common generalization metrics? A decision flowchart or table mapping generalization metrics to recommended calibration metrics would also be helpful for practitioners applying the proposed framework.
-  How could the arbitrariness in selecting kernel types and hyperparameters for calibration-sharpness diagrams be minimized? Could you suggest specific strategies, such as cross-validation approaches or sensitivity analyses, to guide these choices? Additionally, are there recommended default choices for kernels and hyperparameters that perform robustly across various scenarios?
- Given the variability in calibration performance across different kernel types and hyperparameter settings (as shown in the Appendix), could you provide more guidance on appropriate criteria for kernel selection and hyperparameter tuning? This would strengthen the reliability of calibration assessments based on the proposed diagrams.
- Could you clarify the relationship between some theorems and the main discussion, e.g.,  regarding the “always wrong predictor” in Proposition 4.5 and its implications? An example or illustration of how the “always wrong predictor” relates to real-world calibration scenarios would be helpful. Additionally, could you explain how this proposition supports or challenges the main arguments of the paper?
- Could you expand on the motivation behind choosing Bregman divergences for calibration metrics? Specifically, what factors led to the choice of Bregman divergences over other potential metrics? Additional context on this decision-making process would be helpful.
  - Related to the above question, would you consider adding a brief explanation of Bregman divergences and their relationship to calibration metrics in Section 2? I believe this might improve the flow from Section 3 to Section 4 and provide readers with a clearer understanding of the theoretical foundation.
- I may have overlooked or misunderstood certain aspects, but could you provide a more detailed guide on how to interpret the components of experimental figures like Figure 1? Specifically, explanations on how to read the sharpness band, density plot, and calibration curve, and what each represents in terms of model performance, would be helpful. Adding some sentences in the caption part of the figures or including a “How to read this figure” section in the paper could also improve accessibility.
- Although it might slightly deviate from the core contributions of this paper, would it be valuable to provide empirical discussions on the bias of the Nadaraya-Watson estimator? For instance, as demonstrated in [Jiang et al., ICML 2020], would it be possible to use a synthetic data setup to empirically evaluate the bias of the Nadaraya-Watson estimator by estimating the true pointwise sharpness gap with Monte Carlo estimation and comparing it to the Nadaraya-Watson estimator? If this is not feasible within the setup of [Jiang et al., ICML 2020], could a simpler experimental setting be devised where the true values of the pointwise sharpness gap are estimable?

### Citation:
[Jiang et al., ICML 2020]: Jize Zhang, Bhavya Kailkhura, and T. Yong-Jin Han. Mix-n-Match: Ensemble and Compositional Methods for Uncertainty Calibration in Deep Learning. ICML2020. (https://arxiv.org/abs/2003.07329).

**Details Of Ethics Concerns:**

N/A.

---

> ### Author Response · Authors · 2024-11-15
> **Response to Reviewer sELH (Part 1)**
>
> We would like to thank Reviewer sELH for taking the time to review our paper. We are grateful that they found the insights in our paper to be valuable, and we hope to address the pointed out questions and concerns below.
>
> > Could you provide specific examples or a more detailed workflow for selecting calibration metrics based on common generalization metrics? A decision flowchart or table mapping generalization metrics to recommended calibration metrics would also be helpful for practitioners applying the proposed framework.
>
> We have revised the text in Section 4 to make it clearer how to practically apply our framework. In particular, we emphasize that all one has to do to obtain a notion of calibration error is to take the generalization error (which is applied to labels $Y$ and model predictions $g(X)$) and then apply that error to the conditional expectation $\mathbb{E}[Y \mid g(X)]$ and the predictions $g(X)$. In this sense there is no need for a table/flowchart, as one can obtain infinitely many different choices of calibration error this way (once we fix our generalization error). Furthermore, most of these calibration errors are not "named" (like ECE) since they are derived directly from the generalization error -- it just happens that the $L^2$ ECE is derived from Brier score.
>
> > How could the arbitrariness in selecting kernel types and hyperparameters for calibration-sharpness diagrams be minimized? Could you suggest specific strategies, such as cross-validation approaches or sensitivity analyses, to guide these choices? Additionally, are there recommended default choices for kernels and hyperparameters that perform robustly across various scenarios?
>
> For selecting kernels/kernel hyperparameters, you are absolutely right; the best way to do this would be to have a downstream task/comparison and then tune them with respect to some held-out validation data from that task. We now describe this approach in Appendix H. The tuning of these hyperparameters is trickier when the main product of the method is a visualization, but we can still tune using the estimated calibration error values since those are also produced using the same kernel/kernel bandwidth. As far as recommended defaults are considered, Gaussian and Epanechnikov kernels are standard choices - as we discuss the next point, it was previously investigated in [1] how results actually do not change much as we vary the bandwidth for these choices beyond a certain point.
>
> > Given the variability in calibration performance across different kernel types and hyperparameter settings (as shown in the Appendix), could you provide more guidance on appropriate criteria for kernel selection and hyperparameter tuning? This would strengthen the reliability of calibration assessments based on the proposed diagrams.
>
> Although there is variability in the visualizations across different kernels and hyperparameters, we actually interpret the results of Appendix H as showing that for reasonable kernels/hyperparameters the relationships between different model performances are preserved. We understand here that "reasonable" is not well-defined; we now provide more justification in the form of referencing prior work in Appendix H. Historically, people have most frequently compared calibration error using binning estimators with the number of bins being in the range 15-20; while this seems arbitrary, the experiments of [1] show that reported ECE results don't really change as we increase the number of bins beyond this point (but they can change significantly when we decrease the number of bins too much). [1] also provides some explanation for why we can consider taking the bandwidth to be 1/number of bins.
>
> [1] Chidambaram, M., Lee, H., McSwiggen, C., & Rezchikov, S. (2024). How Flawed is ECE? An Analysis via Logit Smoothing. ArXiv, abs/2402.10046.

---

> > ### Author Response · Authors · 2024-11-15
> > **Response to Reviewer sELH (Part 2)**
> >
> > > Could you clarify the relationship between some theorems and the main discussion, e.g., regarding the “always wrong predictor” in Proposition 4.5 and its implications? An example or illustration of how the “always wrong predictor” relates to real-world calibration scenarios would be helpful. Additionally, could you explain how this proposition supports or challenges the main arguments of the paper?
> >
> > Proposition 4.5 is just meant to indicate the theoretical perils of applying the confidence calibration formulation entirely to the multi-class problem (i.e. instead of computing generalization error with respect to $Y$ and $g(X)$, we compute it with respect to $1_{Y = c(X)}$ and $h(X)$). Basically, we show that the full generalization error can be nontrivially large while the reduced/confidence version of the generalization error can be vanishingly small. We don't mean to claim that this has actually been a major problem in practice, but since there is at least one recent prior work that has reported error in this way we wanted to point it out as something to watch out for. This proposition also serves to motivate our framework of reporting the full multi-class generalization error but using the reduced/confidence calibration error, since that gives us the best of both worlds (both quantities are efficient to estimate).
> >
> > > Could you expand on the motivation behind choosing Bregman divergences for calibration metrics? Specifically, what factors led to the choice of Bregman divergences over other potential metrics? Additional context on this decision-making process would be helpful.
> >
> > In the revision, we have added more text in Section 4 to indicate why we choose to focus on Bregman divergences. As you correctly point out, Bregman divergences are a general enough notion that capture various generalization errors of interest while still having a sufficient amount of theoretical structure to prove interesting and useful results. For example, Lemma 4.6 is a key result in our paper and relies strongly on the fact that Bregman divergences are induced by convex functions.
> >
> > > I may have overlooked or misunderstood certain aspects, but could you provide a more detailed guide on how to interpret the components of experimental figures like Figure 1? Specifically, explanations on how to read the sharpness band, density plot, and calibration curve, and what each represents in terms of model performance, would be helpful. Adding some sentences in the caption part of the figures or including a “How to read this figure” section in the paper could also improve accessibility.
> >
> > We apologize for not clarifying appropriately how to read our calibration-sharpness diagrams in the initial version of our submission, and thank you for this important suggestion. In the revision, we have now included a description under "Constructing and reading calibration-sharpness diagrams" in Section 4.2 that goes over each component of the diagram and how to interpret them.
> >
> > > Although it might slightly deviate from the core contributions of this paper, would it be valuable to provide empirical discussions on the bias of the Nadaraya-Watson estimator? For instance, as demonstrated in [Jiang et al., ICML 2020], would it be possible to use a synthetic data setup to empirically evaluate the bias of the Nadaraya-Watson estimator by estimating the true pointwise sharpness gap with Monte Carlo estimation and comparing it to the Nadaraya-Watson estimator? If this is not feasible within the setup of [Jiang et al., ICML 2020], could a simpler experimental setting be devised where the true values of the pointwise sharpness gap are estimable?
> >
> > In the revision, we now provide more discussion regarding the use of kernel regression estimators for the conditional expectation in Appendix D; a complete discussion would be out of the scope of this work but we now reference the appropriate texts containing the key results on kernel regression asymptotic properties (bias, consistency, convergence rates).
> >
> > We are happy to address any follow-up concerns you may have on top of the above answers and revisions we have made to the paper (see top-level comment for a full description).

---

> > > ### Comment · Reviewer_sELH · 2024-11-22
> > > **Reply for Response**
> > >
> > > Thank you sincerely for addressing my concerns so thoroughly. First, I would like to apologize for my partial misunderstanding regarding the workflow of the generalized metric.
> > >
> > > Assuming that the content of your responses is reflected in the manuscript, I believe this paper meets the quality standards required for acceptance at ICLR. The contributions and insights are both interesting, and it is a solid piece of work.
> > >
> > > Therefore, I would like to update my score to 6.

---

### Official Review · Reviewer_6vrr · 2024-11-04

**Soundness:** 3
**Presentation:** 3
**Contribution:** 2
**Rating:** 6
**Confidence:** 2

**Summary:**

The paper critically examines current practices in evaluating model calibration, particularly for deep learning. It highlights flaws in reporting calibration metrics. The authors propose a new methodology based on the calibration-based decomposition of Bregman divergences and introduce new diagrams. These diagrams visualize calibration and generalization error jointly, allowing a more nuanced assessment. Additionally, the paper provides theoretical insights.

**Strengths:**

1. This paper studies a fundamental question in the ML calibration literature: how to compare and assess the model calibration performance. This is a very important question.
2. This paper is easy to follow, and the main take-away is very clear.
3. This paper points out "pitfalls" regarding calibration reporting in the current literature clearly.
4. The proposed calibration-sharpness digram is interesting and novel.
5. This paper is theoretically sound.

**Weaknesses:**

1. The experiment section is relatively weak. Only one dataset was examined.
2. It would be great if the authors could open-source the code and make the new diagrams easier to use for other ML practitioners.

**Questions:**

n/a

---

> ### Author Response · Authors · 2024-11-15
> **Response to Reviewer 6vrr**
>
> We would like to thank Reviewer 6vrr for reviewing our paper, and we are glad they found the paper well-presented and the visualization interesting and novel. We hope to address the stated weaknesses below.
>
> > 1. The experiment section is relatively weak. Only one dataset was examined.
>
> We have now included a Subsection in the Appendix (Subsection F.2) that includes experiments on CIFAR-10 and CIFAR-100; these experiments further corroborate our findings in the main paper.
>
> > 2. It would be great if the authors could open-source the code and make the new diagrams easier to use for other ML practitioners.
>
> In the original submission of the paper, we included code in the supplementary material that provides all the utilities necessary to recreate our results. After the paper is deanonymized, we will link to an open source version of this code directly in the paper.
>
> We are happy to address any further concerns you may have. We hope that the above, along with the other revisions we describe in the top-level comment, serve to better highlight the usefulness/novelty of our results.

---

> ### Comment · Reviewer_6vrr · 2024-11-25
>
> Thanks for the reply. I will maintain my score.

---

### Author Response · Authors · 2024-11-15
**Overview of Revisions**

We would like to thank all of the reviewers for their helpful suggestions. Here we outline the main changes we have made in the revised version of the submission.

1. **Clearer description of calibration-sharpness diagrams.** We now have a "constructing and reading calibration-sharpness diagrams" part of Section 4.2, which hopefully significantly clarifies how to interpret the diagrams. More fine-grained details regarding the visualization and estimation approach have been moved to Appendix D.

2. **Better comparison between reliability diagrams and calibration-sharpness diagrams.** We have revised Section 5.2 to show a more telling comparison between standard reliability diagrams and calibration-sharpness diagrams, and have also provided a brief description of reliability diagrams. Furthermore, we have also included a comparison to SmoothECE [1] diagrams in Appendix G. We hope these comparisons better highlight the benefits of our proposed approach.

3. **Additional experiments.** We have now included experiments on CIFAR-10 and CIFAR-100 in Appendix F.2 to further corroborate our findings on ImageNet.

4. **Expanded related work.** Appendix E now carefully compares our work to recent work on analyzing the grouping loss, which we hope further contextualizes our results.

5. **Cleaned up proofs.** We have cleaned up the proofs in the appendix, fixing minor typos, adding detail where it was previously missing, and also improving the organization of the results.

6. **Better motivation for working with Bregman divergences.** We have revised the writing in Section 4 to better explain why we work with Bregman divergences and also how one should think about applying our framework.

We hope to continue to engage with the reviewers and improve our paper.

[1] Jaroslaw Blasiok and Preetum Nakkiran. Smooth ece: Principled reliability diagrams via kernel smoothing, 2023.

---

### Meta-Review · Area_Chair_3Tu5 · 2024-12-17

**Metareview:**

This work proposes new methods for evaluating calibration in ML. The authors were able to address all issues raised by the reviewers. All reviewers except one were positive about the work. The authors prepared extensive updates and addressed the issues raised by this last reviewer, who however was not able to respond. Since all issues have been resolved, I believe the paper would be a valuable contribution.

**Additional Comments On Reviewer Discussion:**

During the rebuttal,  the authors clarified theoretical contributions, improved plot interpretability, and expanded experiments. They added a detailed guide to understanding calibration-sharpness diagrams and better motivated their use of Bregman divergences. The authors introduced additional comparisons to alternative methods, such as SmoothECE, and improved the consistency and detail of proofs. They also provided a comprehensive discussion on kernel and hyperparameter selection, referencing prior work to support their methodology. Furthermore, the authors included additional experiments.

---

### Decision · Program_Chairs · 2025-01-22

Accept (Poster)